# A high-throughput screen indicates gemcitabine and JAK inhibitors may be useful for treating pediatric AML

Christina D. Drenberg[1,2], Anang Shelat [3], Jinjun Dang[4], Anitria Cotton[4], Shelley J. Orwick[2], Mengyu Li[1,2], Jae Yoon Jeon[1,2], Qiang Fu[1,2], Daelynn R. Buelow[1,2], Marissa Pioso[1,2], Shuiying Hu[1,2], Hiroto Inaba[4], Raul C. Ribeiro[4], Jeffrey E. Rubnitz[4], Tanja A. Gruber[4], R. Kiplin Guy [5] & Sharyn D. Baker[1,2]

Improvement in survival has been achieved for children and adolescents with AML but is largely attributed to enhanced supportive care as opposed to the development of better treatment regimens. High risk subtypes continue to have poor outcomes with event free survival rates <40% despite the use of high intensity chemotherapy in combination with hematopoietic stem cell transplant. Here we combine high-throughput screening, intracellular accumulation assays, and in vivo efficacy studies to identify therapeutic strategies for pediatric AML. We report therapeutics not currently used to treat AML, gemcitabine and cabazitaxel, have broad anti-leukemic activity across subtypes and are more effective relative to the AML standard of care, cytarabine, both in vitro and in vivo. JAK inhibitors are selective for acute megakaryoblastic leukemia and significantly prolong survival in multiple preclinical models. Our approach provides advances in the development of treatment strategies for pediatric AML.

[1] Division of Pharmaceutics and Pharmaceutical Chemistry, College of Pharmacy, The Ohio State University, Columbus, OH 43210, USA. [2] Comprehensive Cancer Center, The Ohio State University, Columbus, OH 43210, USA. [3] Department of Chemical Biology and Therapeutics, St. Jude Children's Research Hospital, Memphis, TN 38105, USA. [4] Department of Oncology, St. Jude Children's Research Hospital, Memphis, TN 38105, USA. [5] Department of Pharmaceutical Sciences, College of Pharmacy, University of Kentucky, Lexington, KY 40506, USA. Correspondence and requests for materials should be addressed to C.D.D. (email: guttke.1@osu.edu)

Dramatic improvements in survival have been achieved for children and adolescents with acute myeloid leukemia (AML), with 5-year survival rates increasing between 1975 and 2010 from <20% to >70%[1]. These improvements are attributable to the intensification of chemotherapy, selective use of hematopoietic stem cell transplantation, improvements in supportive care, refinements in risk classification, and use of minimal residual disease to monitor response to therapy[2]. However, in the past decade outcome has plateaued and remain unacceptably low for many subtypes. Provided that significant improvements in long-term outcome are not expected with conventional therapy alone, therapeutic strategies that can be quickly advanced to a clinical setting are urgently needed for the treatment of pediatric AML.

Drug discovery and development is a long process that requires an enormous financial investment and multiple clinical trials; a process that has an increased number of challenges and barriers in orphan diseases such as pediatric AML. A more rational, evidence-based approach to identify and prioritize therapeutics advancing to clinical trials is needed. Here, we report the results of a large-scale screen of human cancer cell lines representing two high-risk subtypes of pediatric AML including MLL rearranged (MLLr) with or without a co-occurring FLT3-internal tandem duplication (FLT3-ITD) mutation and non-Down syndrome acute megakaryoblastic leukemia (AMKL)[2–4]. Both FLT3-ITD and MLLr occur in adult AML patients and are associated with poor prognosis whereas, AMKL is extremely rare occurring in only 1% of the adult patient population[5–7]. This provides a strong rationale for identifying treatment strategies that may be unique to pediatric AML and those that may have broader implications. The Broad and Sanger Institutes independently reported the results from two large-scale screens, which systematically interrogated hundreds of cell lines both genetically and pharmacologically[8,9]. AML represented a very small fraction of cancer types evaluated in these studies and high-risk subtypes including MLLr AML and non-DS AMKL were not represented and the number of compounds evaluated were limited (Broad study, $N = 24$; Sanger study, $N = 130$). Our current study evaluates the anti-leukemic activity of nearly 8000 compounds at a single concentration in a primary high-throughput screen (HTS) using a panel of cell lines derived from children and young adults. We evaluate select compounds representing a variety of drug classes including cytotoxic and molecularly targeted agents in a secondary dose-response HTS. To accelerate the transition of potential therapeutics into the clinic, we prioritize hits based on FDA-approved agents with oncology indications. Anti-leukemic activity is validated in low-throughput assays in vitro with cell lines, ex vivo with primary pediatric patient samples, and in vivo using murine models. This integrated approach identifies two FDA-approved drugs, gemcitabine and cabazitaxel, as therapeutics with broad activity across multiple subtypes of pediatric AML with dismal prognosis. In addition, we report the selective activity of JAK inhibitors for AMKL; specifically the FDA-approved therapeutic agent, ruxolitinib, prolongs survival in multiple murine models of AMKL.

## Results

**Drug screen using AML cell lines**. We validated a panel of 8 AML cell lines derived from children and young adults for screening (Fig. 1a, Supplementary Fig. 1, Supplementary Table 1). A primary HTS was performed using a library of 7389 compounds (6568 unique) at a single concentration. Percent inhibition of cell proliferation was determined relative to the positive control (Supplemental Data 1). Assay diagnostics were acceptable and the scatter-plot of controls and test compound activity demonstrated adequate separation between signal and noise for each cell line (Supplementary Fig. 2). The range of total hits (activity >50% inhibition) in the primary screen was 334–624 and the number of selective hits (activity >80% in one cell line and <20% in all others) was negligible (range 0–6; mean 2.6) (Supplementary Table 2). A secondary HTS performed in a dose-response manner, included FDA approved compounds with inhibition >50% in more than one cell line in the primary screen, analogs of these hits, and other compounds of interest (e.g., NAMPT inhibitors) not included in the primary screen; clinical phase of testing was also taken into consideration. The average effective concentration of each compound is reported in Fig. 1b and Supplementary Data 2. Of the 458 compounds, we identified 17 with potency <1 μM in all cell lines; collectively, these included histone deacetylase inhibitors (5/17), proteasome inhibitors (4/17), PI3K inhibitor (1/17), inhibitors of anti-apoptotic proteins (2/17), and FDA-approved cytotoxic agents (5/17) two of which (clofarabine, mitoxantrone) are currently used in clinical regimens for the treatment of AML (Table 1). We validated compounds from these drug classes and others targeting pathways known to be upregulated or mutated in MLLr or AMKL (e.g., trametinib, RAS pathway; alisertib, aurora kinase; RG7112, MDM2 inhibitor)[10–12] using cell lines and primary patient samples (Fig. 2, Supplementary Fig. 3, Supplementary Table 3). Primary patient samples were co-cultured with mesenchymal stromal cells, which secrete multiple cytokines that mimic the bone marrow microenvironment; this system gives support to primary samples while challenging the drug treatment. Cell viability and cell density were monitored throughout the assay (Supplementary Fig. 4); these data demonstrate all primary samples experience a dramatic decrease in cell number at 24 h; though cell numbers are relatively maintained over the course of the assay only one sample doubles from 24 to 96 h. This is an important observation especially in regard to drugs that specifically target S phase cells and may contribute to the modest activity of nucleoside analogues like cytarabine and gemcitabine in this assay. We observed the HDAC inhibitors panobinostat and romidepsin to have potent activity across subtypes; these findings are consistent with our previous report with panobinostat[13] and support the ongoing clinical evaluation (NCT02676323) of this drug for pediatric AML. Similarly, the proteasome inhibitors, carfilzomib and bortezomib, demonstrated potent activity and have been extensively investigated in the clinic[14]. Criterion for the advancement of compounds are described in the Methods section and may permit rapid translation to a clinical trial in pediatric AML; this is essential as we currently have access to few if any investigational agents.

**Gemcitabine demonstrates potent in vitro activity**. Given that nucleoside analogs are integral to all modern AML therapy and since gemcitabine, a nucleoside analog that is currently used to treat advanced solid tumors in children[15,16] demonstrated very potent activity across subtypes in the secondary HTS (<65 nM); and taken these results were validated in a low-throughput manner (Fig. 3a) and had comparable activity to cytarabine in primary patient samples (Fig. 2, Supplementary Table 3), we selected this compound for further evaluation. A panel of AML cell lines were treated with increasing concentrations of gemcitabine and clinically used nucleoside analogs for comparison. Both cytarabine and fludarabine demonstrated variable activity; whereas both clofarabine and gemcitabine had more narrow half maximal inhibitory concentration (IC$_{50}$) ranges (Fig. 3a). Overall, gemcitabine was the most potent and had activity in cell lines that were insensitive to cytarabine (THP-1) and fludarabine (CHRF288-11, MV4-11).

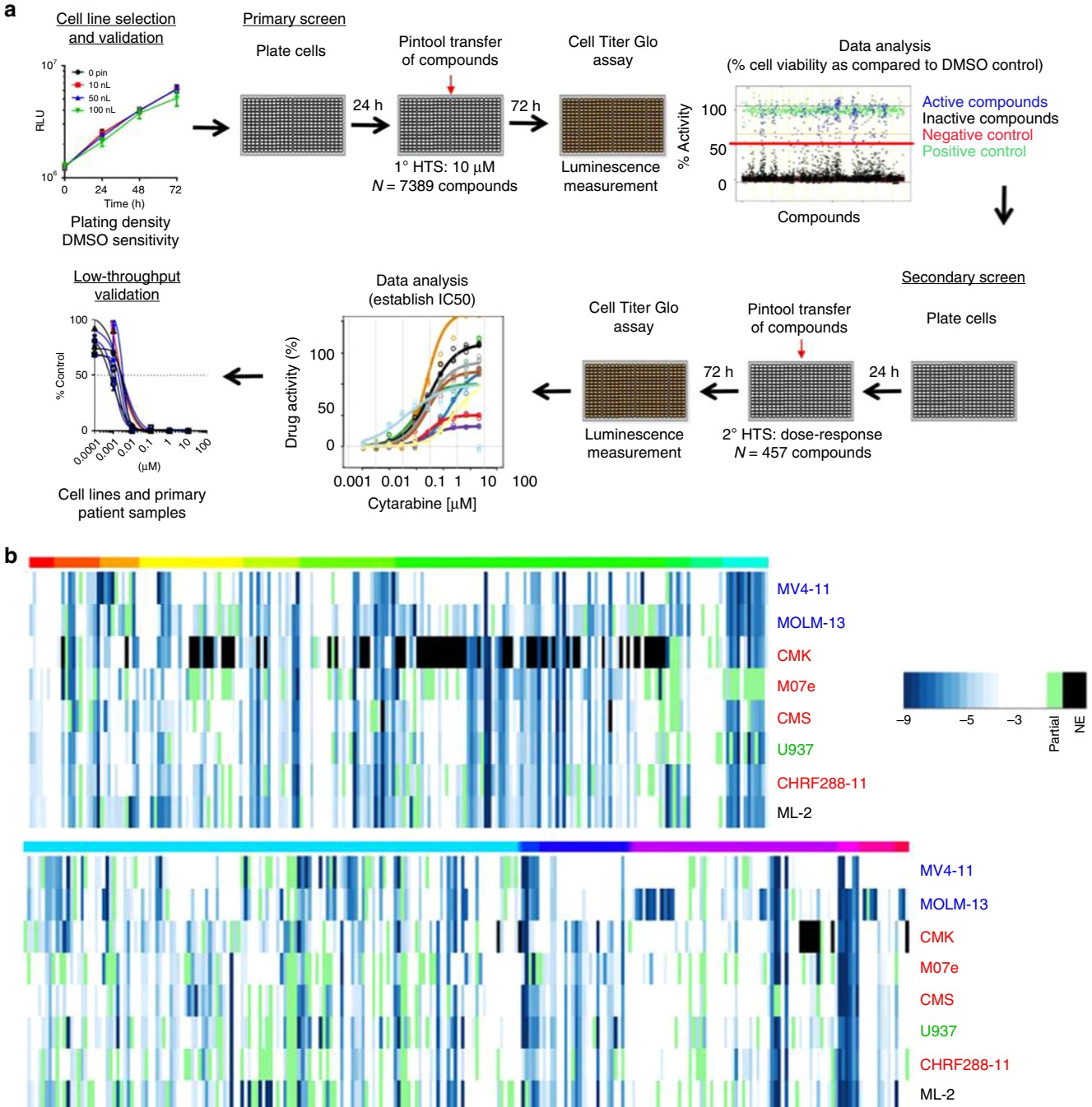

**Fig. 1** High-throughput screening of pediatric AML. **a** Illustration of the scheme used for AML cells in the screening platform. Optimal plating density was determined per cell line in a 384-well plate; a primary screen was conducted at a single concentration and a secondary screen was performed in a dose-response manner in triplicate. For screens, cells were plated and after 24 h compounds were pin tool transferred using an automation station. Cell viability was measured at 72 h using Cell Titer Glo. Select compounds were validated using cell lines, primary patient samples, and/or in vivo murine models. **b** Heatmap of the average effective concentration ($EC_{50}$) from secondary screen. Cell lines are ordered based on a cluster analysis. Black, *MLL* rearranged (*MLLr*); blue, FLT3-internal tandem duplication positive with *MLLr*; green, *PICALM/MLLT10* fusion positive; red, acute megakaryoblastic leukemia. Color bar on top of heatmap indicates compound classes: red, anti-infective and anti-psychotic; orange-red, anti-metabolite; orange, apoptosis; yellow, DNA damage; lime, complex; green, folate, epigenetic, retinoic acid receptor; teal, Hsp90; light blue, kinase; blue, microtubule, NF-κB; purple, other; light pink, proteasome; pink, HIF, Nrf2; NE, not evaluated

Collectively, nucleoside analogs share a similar mechanism of action whereby they enter cells exclusively by transporter-mediated processes[17]. Upon intracellular uptake, multiple rounds of phosphorylation must occur before insertion into DNA[18]. Reduced uptake into leukemia cells has been proposed as a process underlying most instances of clinical resistance to cytarabine, though the responsible mechanism remains poorly understood[19,20]. To investigate whether differences in cellular uptake were contributing to the enhanced anti-leukemic activity of gemcitabine, we performed intracellular uptake and accumulation experiments. In a comparative analysis after 5 min exposure to cytarabine or gemcitabine, we detected a significantly greater

**Table 1 Compounds with EC$_{50}$ < 1 μM in all cell lines evaluated in secondary HTS**

| Compound | MOLM-13 | MV4-11 | CHRF288-11 | CMK | CMS | M07e | ML2 | U937 |
|---|---|---|---|---|---|---|---|---|
| Clofarabine[a] | 0.05 | 0.15 | 0.37 | 0.005 | 0.02 | 0.01 | 0.13 | 0.09 |
| Gemcitabine[a] | 0.003 | 0.002 | 0.003 | 0.01 | 0.03 | 0.06 | 0.002 | 0.01 |
| Gambogic acid[c] | 0.19 | 0.17 | 0.22 | 0.36 | 0.26 | 0.24 | 0.15 | 0.19 |
| Dactinomycin[d] | 0.03 | 0.04 | 0.001 | 0.001 | 0.04 | 0.001 | 0.001 | 0.001 |
| Mitoxantrone[c] | 0.01 | 0.001 | 0.02 | 0.04 | 0.08 | 0.03 | 0.24 | 0.03 |
| Trichostatin A[e] | 0.08 | 0.04 | 0.04 | 0.05 | 0.04 | 0.09 | 0.11 | 0.04 |
| Quisinostat[e] | 0.01 | 0.01 | 0.01 | 0.02 | 0.001 | 0.004 | 0.004 | 0.01 |
| CUDC-907[e,k] | 0.0004 | 0.0002 | 0.001 | 0.002 | 0.001 | 0.001 | 0.002 | 0.004 |
| Panobinostat[e] | 0.18 | 0.13 | 0.01 | 0.003 | 0.03 | 0.005 | 0.01 | 0.01 |
| Romidepsin[e] | 0.001 | 0.001 | 0.01 | 0.01 | 0.001 | 0.002 | 0.004 | 0.003 |
| NVP-BGT226[k] | 0.11 | 0.05 | 0.05 | 0.06 | 0.01 | 0.25 | 0.23 | 0.08 |
| Cabazitaxel[m] | 0.001 | 0.0003 | 0.001 | 0.001 | 0.001 | 0.001 | 0.001 | 0.001 |
| Ouabain[o] | 0.03 | 0.04 | 0.04 | 0.18 | 0.05 | 0.06 | 0.03 | 0.04 |
| Bortezomib[p] | 0.002 | 0.002 | 0.002 | 0.01 | 0.001 | 0.001 | 0.001 | 0.02 |
| Oprozomib[p] | 0.01 | 0.01 | 0.03 | 0.01 | 0.001 | 0.01 | 0.01 | 0.01 |
| ONX-0914[p] | 0.07 | 0.05 | 0.17 | 0.23 | 0.11 | 0.08 | 0.04 | 0.13 |
| Carfilzomib[p] | 0.01 | 0.001 | 0.02 | 0.01 | 0.001 | 0.001 | 0.003 | 0.01 |

Mechanism of action indication as follows: a—anti-metabolite; c—complex; d—DNA; e—epigenetic; k—kinase; m—mitotic; o—other; p—proteasome

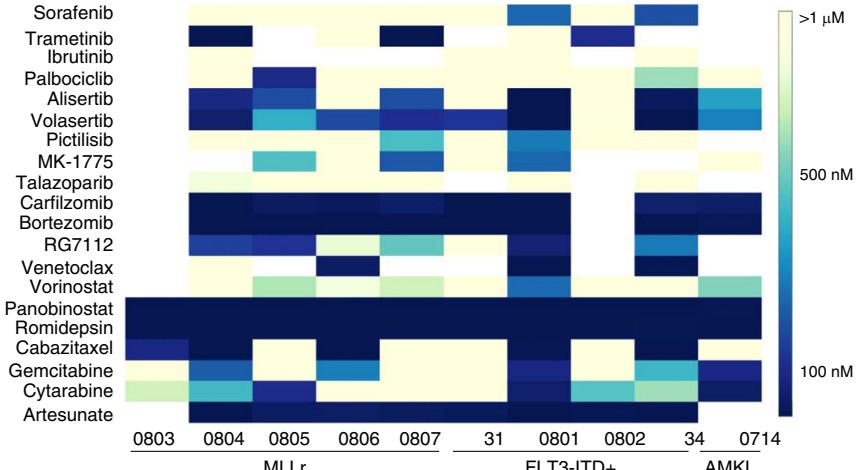

**Fig. 2** Activity of compounds in primary AML blast samples. Primary blast samples from patients harboring a *MLL* rearrangement (*MLLr*), FLT3-internal tandem duplication positive (FLT3-ITD+), or with acute megakaryoblastic leukemia (AMKL) were plated on mesenchymal stromal cells and treated with DMSO or increasing concentrations of indicated compounds. Cell viability was determined at 96 h using Cell Titer Glo. The half maximal inhibitory concentration (IC$_{50}$) was evaluated by nonlinear regression analysis using GraphPad Prism (*N* = 3 per concentration). Heatmap indicates IC$_{50}$ concentration for each compound per patient sample

amount (1.2–3.6-fold) of gemcitabine in all cell lines evaluated (Fig. 3b). This observation continued over a 2-h time course and was associated with significantly increased gemcitabine accumulation (5–19.8-fold compared to cytarabine) in the nuclear compartment (Fig. 3c, d). While the observed nuclear uptake suggests that the active metabolite gemcitabine triphosphate (dFdCTP) was accumulating to a much greater extent relative to cytarabine triphosphate (Ara-CTP); these studies detect total radioactivity, which is comprised of both active and inactive metabolites. To determine if we were observing greater accumulation of dFdCTP, we evaluated the accumulation of gemcitabine monophosphate (dFdCMP), diphosphate (dFdCDP), and dFdCTP compared to the metabolic counterparts of cytarabine (Ara-CMP, Ara-CDP, Ara-CTP). After 2 h exposure, significantly greater accumulation of dFdCTP compared to Ara-CTP (7.3–61.2-fold) was observed; dFdCTP accounted for 58–81% of the total intracellular accumulation in all cell lines

evaluated whereas Ara-CTP accounted for 52–65% (Fig. 3e, Supplementary Fig. 5a–c). Additionally, we evaluated the accumulation using primary murine blasts isolated from the bone marrow and spleen of treatment-naive Mll$^{PTD/wt}$:Flt3$^{ITD/ITD}$ double knock-in primary transplants. Similarly, we observed higher accumulation of total gemcitabine versus cytarabine (6.0–6.7-fold) and dFdCTP versus Ara-CTP (2.4–5.3-fold) in blasts from bone marrow or spleen (Supplementary Fig. 5d).

To determine if dose escalation could result in equivalent intracellular exposure, we performed uptake assays at higher concentrations of cytarabine. After 5 min exposure to cytarabine (10 or 100 μM) we detected a significantly greater amount (1.6–3.7-fold and 7–12-fold, respectively) compared to 1 μM cytarabine (Fig. 3f). Furthermore, we found a 10-fold higher concentration of cytarabine produced nearly equivalent intracellular exposure to 1 μM gemcitabine. Since transporters are biophysically complex and the interpretation of

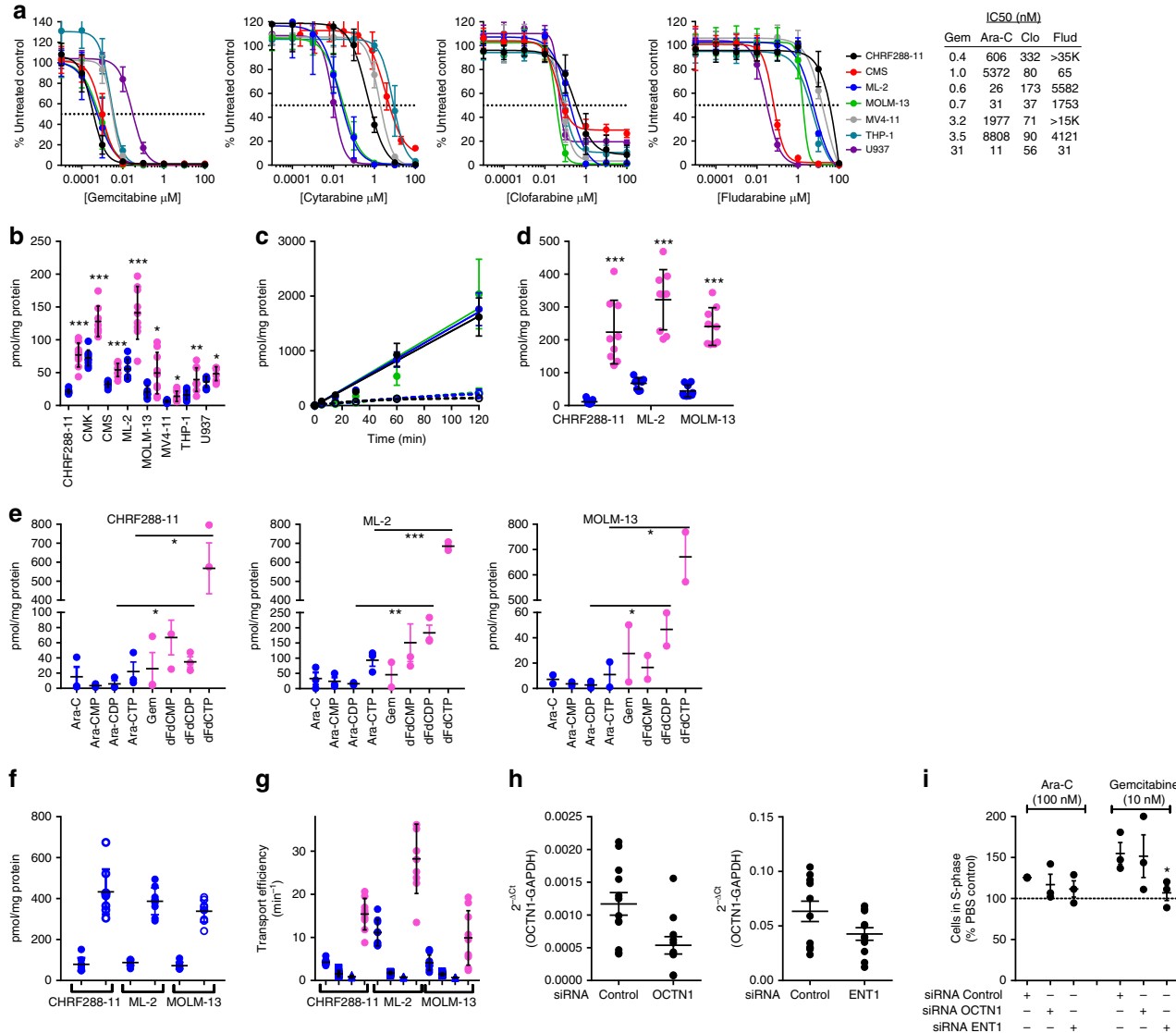

**Fig. 3** Activity of gemcitabine in AML cell lines. **a** AML cells were treated with increasing concentrations of vehicle or nucleoside analog (gemcitabine, Gem; cytarabine, Ara-C; clofarabine, Clo; and fludarabine, Flud) for 72 h, cell viability was measured using Cell Titer Glo. Data are reported as percent control and represented as mean ± standard deviation (SD) of 3 independent experiments ($N = 18$ per concentration) The half maximal inhibitory concentration (IC$_{50}$, dotted line) was evaluated by nonlinear regression analysis using GraphPad Prism. Uptake/accumulation assays used a mixture of unlabeled and [3H]-Ara-C (blue circles) or Gem (magenta circles). **b** Intracellular uptake (5 min, 1 µM) of Ara-C and Gem. Data are mean ± SD from 3 independent experiments performed in triplicate ($N = 9$). **c** Time-dependent accumulation (range, 5–120 min; 1 µM) of Ara-C (dashed line) or Gem (solid line) in MOLM-13 (magenta), ML-2 (blue), and CHRF288–11 (black). Data are mean ± SD from 3 independent experiments performed in triplicate ($N = 9$). **d** Nuclear accumulation (2 h, 1 µM) of Ara-C and Gem. Data are mean ± SD from 3 independent experiments performed in triplicate ($N = 9$). **e** Intracellular Ara-C, Gem, and phosphorylated metabolites (Ara-CMP, Ara-C monophosphate; Ara-CDP, Ara-C diphosphate; Ara-CTP, Ara-C triphosphate; dFdCMP, Gem monophosphate; dFdCDP, Gem diphosphate; dFdCTP; Gem triphosphate) was determined by HPLC coupled with liquid scintillation counting (2 h, 1 µM). Data are mean ± standard error (SE) from 3 independent experiments ($N = 3$). **f** Uptake (5 min) of 10 µM (solid blue circles) or 100 µM (open blue circles) Ara-C. Data are mean ± SD from 3 independent experiments performed in triplicate ($N = 9$). **g** Transport efficiency was determined for uptake (5 min) assays for Ara-C (blue; circle, 1 µM; square, 10 µM; triangle, 100 µM) and Gem (magenta circle, 1 µM). Data are mean ± SD from 3 independent experiments ($N = 9$). **h** Confirmation of target knockdown in MOLM-13 cells at 48 h after transient transfection with siRNA-Control, -OCTN1, or -ENT1; data are mean ± SE from 4 independent experiments performed in triplicate ($N = 12$). **i** Cytotoxicity was assessed following 16 h exposure to 100 nM Ara-C or 10 nM Gem by induction of S-phase of the cell cycle in transfected cells. Data are mean ± SE from 3 independent experiments ($N = 3$). Student's *t* test. *$P < 0.05$; **$P < 0.01$; ***$P < 0.0001$

Michaelis–Menten parameters with regard to underlying transporter dynamics can be difficult[21–23], we evaluated transporter kinetics by calculating the transport efficiency for these compounds to gain a greater mechanistic understanding of these data. Overall transport efficiency of gemcitabine was significantly greater compared to cytarabine and this could not be overcome with increasing concentrations (Fig. 3g).

Given that previous investigations demonstrated that low cytarabine uptake in AML cells predicts poor response to therapy[17] and taken our data showing uptake and transport efficiency as a major contributor to the enhanced sensitivity of AML cells to gemcitabine we performed knockdown experiments targeting two key transporters involved in nucleoside uptake OCTN1 and ENT1[20,24]. Using siRNA we were able to achieve a

54% and 41% reduction in expression of OCTN1 and ENT1, respectively (Fig. 3h). At 24 h post-transfection, cells were treated with PBS, gemcitabine, or cytarabine then alterations in their cytotoxicity profile were assessed using perturbations in cell cycle as the read-out. Inhibition of both transporters resulted in a decrease of cytarabine- and gemcitabine-mediated accumulation of cells in S-phase; the effect was greater with gemcitabine in ENT1 deficient cells and comparable in both OCTN1 and ENT1 deficient cells with cytarabine (Fig. 3i).

**Taxanes demonstrate potent in vitro activity**. Another class of cytotoxic agents demonstrating potent activity across subtypes were microtubule poisons, specifically taxanes. Cabazitaxel demonstrated potent single agent activity in the secondary HTS and primary patient samples (Figs. 1b and 2, Table 1); likewise, docetaxel demonstrated potent activity in all but one cell line (Supplementary Data 2). These results are consistent with a recent report showing that primary AML cells were sensitive to docetaxel when drug sensitivity and resistance testing was performed in the presence of HS-5 human bone marrow stromal cell conditioned media[25]. We validated the results from the HTS and found cabazitaxel, docetaxel, and paclitaxel have potent activity in a panel of AML cell lines (Fig. 4a); cabazitaxel was the most potent so we selected this compound for further evaluation.

Microtubule poisons like taxanes interfere with the faithful segregation of chromosomes by binding tubulin and disrupting the mitotic spindle causing a G2/M cell cycle arrest, leading to mitotic catastrophe, and ultimately resulting in cell death[26]. Therefore, we assessed the mechanism by which cabazitaxel triggers cell death in AML cells by analyzing cell cycle perturbations and using a biparametric approach of markers associated with mitosis. A time-course analysis showed that cell cycle blockage was detected prior to cell death: G2/M arrest was detectable at 4 h and more prominent at 8 and 12 h, whereas cell death was detectable at 12 h and more pronounced at 24 h (Fig. 4b, Supplementary Fig. 6). The G2/M arrest was accompanied by a significant induction of cyclin B and phosphorylation of histone H3 (Fig. 4c). These results suggest that cabazitaxel induces a mitotic arrest in AML cells, which leads to cell death.

**Establishment of AMKL murine models**. Development and implementation of preclinical in vivo models that faithfully recapitulate human disease are imperative to enhance the predictive power of potential therapeutics. In an effort to establish murine models for AMKL, we labeled CHRF288-11 cells with YFP/luciferase (CHRF288-11-Luc+) to permit monitoring engraftment through bioluminescence imaging; we found CHRF288-11-Luc+ cells could engraft in both female and male mice (Supplementary Fig. 7). We have previously identified a pediatric-specific rearrangement yielding a *CBFA2T3/GLIS2* fusion which is associated with aberrant JAK/STAT signaling and often co-occurs with mutations in *JAK* kinase family members, *STAT* genes, or the thrombopoietin receptor *MPL*[27]; patients with this lesion show the strongest negative association with survival and highest cumulative incidence of relapse or primary resistance[28]. While expression of this gene fusion results in increased self-renewal, transplantation of fusion gene-modified bone marrow cells fails to induce leukemia suggesting there is an essential requirement for cooperating mutation(s) in cases expressing the gene fusion[29,30]. We evaluated the transforming potential of dual expression of *CBFA2T3-GLIS2* with a clinically relevant *JAK2*V617F cooperating mutation (CG/V617); concurrent expression uniformly induced a rapid and fatal leukemia characteristic of AMKL and transplantable into subsequent recipients (Supplementary Fig. 8). We then incorporated luciferase into CG/

V617 AMKL blasts isolated from secondary transplant (CG/V617-Luc+). The bioluminescence signal could be detected in the hind limbs 15 days earlier than CG/V617 cells in the peripheral blood and CG/V617-Luc+ transplants exhibited a highly penetrant phenotypically similar AMKL to the CG/V617 transplants (Supplementary Fig. 8e). We also established a patient-derived xenograft (PDX) using a sample from a pediatric patient with AMKL that was previously identified to carry the *CBFA2T3-GLIS2* fusion plus copy number alterations and amplification on chromosome 21, a major cooperating event that includes genes in the Down Syndrome critical region[27]. These models replicate many features of human AMKL and provide a robust tool set for preclinical evaluation of therapeutic strategies.

**Gemcitabine and cabazitaxel prolong in vivo survival**. Next, we sought to compare the in vivo efficacy of gemcitabine and cabazitaxel to the standard of care, cytarabine. Due to limitations with tolerability, we have previously treated our AML xenograft models with low-dose cytarabine[13,31]. This regimen did not provide any survival advantage compared to vehicle treated mice (median survival 26 versus 26 days) in the CHRF288-11-Luc+ model (Supplementary Fig. 9). Tolerability studies of gemcitabine at the same dose and schedule was not tolerated. Therefore, we performed tolerability using multiple doses on an intermittent every 3 or 4-day schedule; all regimens of gemcitabine were well tolerated. Similarly, we performed tolerability of cabazitaxel using multiple doses on an intermittent every 3 or 4-day schedule; the only tolerable dose was 5 mg/kg.

For efficacy studies performed using immunocompromised mice, we found cytarabine did not provide any survival advantage compared to vehicle treated mice in cell line xenografts; whereas in the AMKL PDX cytarabine significantly prolonged survival (log-rank test, $P = 0.0023$) (Fig. 5, Supplementary Figs. 10–11). In the AMKL xenograft models, gemcitabine provided the greatest survival advantage and significantly prolonged survival versus cytarabine after one (log-rank test, $P = 0.0023$) or two treatment cycles (log-rank test; CHRF288-11-Luc+, $P = 0.0019$; AMKL PDX, $P = 0.0026$). Treatment with gemcitabine significantly inhibited tumor burden compared to cytarabine as indicated by a reduced bioluminescence signal and decreased infiltration in the peripheral blood. While cabazitaxel significantly prolonged survival versus cytarabine after one (log-rank test, $P = 0.0023$) or two cycles (log-rank test, $P = 0.0023$) in the CHRF288-11-Luc + xenograft, we did not observe a significant difference in the AMKL PDX (Fig. 5, Supplementary Fig. 10). Surprisingly, we found the additional treatment cycle of cabazitaxel in the CHRF288-11-Luc+ model did not further prolong survival compared to one cycle. Although gemcitabine significantly prolonged survival versus cytarabine in the ML-2 xenograft model (log-rank test, $P = 0.0027$); cabazitaxel provided the greatest survival advantage (log-rank test, $P = 0.0027$) and significantly inhibited tumor burden compared to cytarabine (Fig. 5c, Supplementary Fig. 11).

Next, we sought to evaluate efficacy in two syngeneic murine models using a maximum tolerated dose (MTD) of cytarabine that better reflects clinical regimens. Tolerability of gemcitabine was performed using multiple doses on an intermittent or daily schedule (Supplementary Fig. 12) to establish the MTD. For efficacy studies in CG-V617-Luc+ quaternary transplants, mice were treated with gemcitabine (MTD daily or intermittent) or cytarabine (daily). We found both dosing regimens of cytarabine provided a survival advantage compared to vehicle treated mice (log-rank test, $P = 0.0091$) (Fig. 5d). However, gemcitabine provided the greatest survival advantage (log-rank test, daily, $P = 0.0091$; intermittent, $P = 0.004$) and significantly prolonged

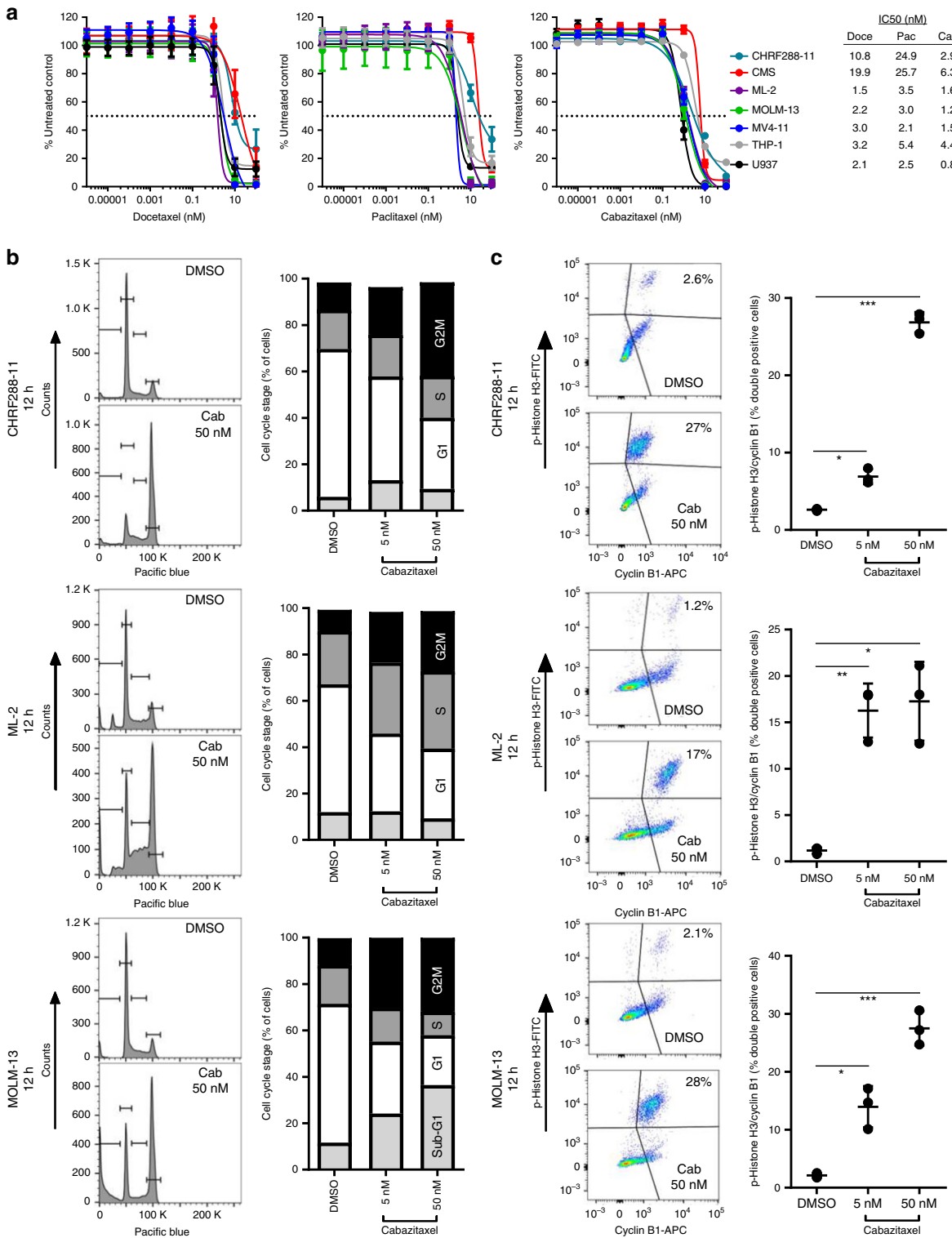

**Fig. 4** Activity of taxanes in AML cell lines. **a** AML cells were treated with increasing concentrations of vehicle or taxane (cabazitaxel, Cab; docetaxel, Doce; paclitaxel, Pac) for 72 h, and cell viability was measured using Cell Titer Glo. Data are reported as percent control and represented as mean ± standard deviation (SD) of 3 independent experiments ($N = 18$ per concentration). The half maximal inhibitory concentration ($IC_{50}$) was evaluated by nonlinear regression analysis using GraphPad Prism. CHRF288-11, ML-2, and MOLM-13 cells were treated with 5 or 50 nM Cab or DMSO for 12 h; **b** cell cycle distribution was determined by DAPI staining and **c** induction of mitosis was determined by biparametric flow cytometry using p-histone H3 and cyclin B1. Data are mean ± standard deviation from 3 independent experiments ($N = 3$); Student's $t$ test. *$P < 0.01$; **$P < 0.001$; ***$P < 0.0001$

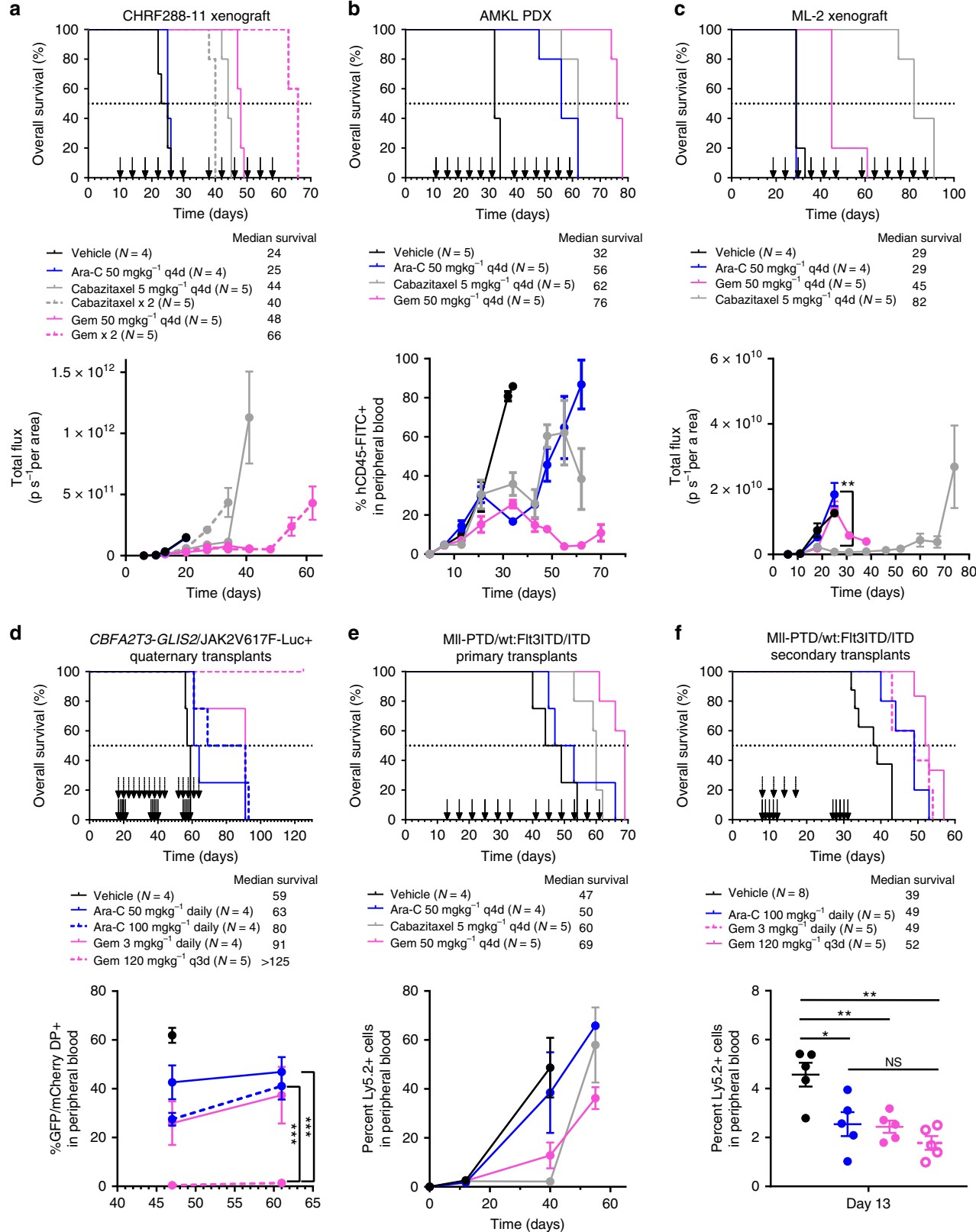

survival versus cytarabine (log-rank test, $P = 0.0027$). Treatment with gemcitabine significantly inhibited tumor burden compared to cytarabine as indicated by decreased infiltration in the peripheral blood (Fig. 5d).

Lastly, we evaluated efficacy using a $Mll^{PTD/wt}$:$Flt3^{ITD/ITD}$ double knock-in murine model[32,33]. Primary transplants were

treated with intermittent dosing of gemcitabine, cytarabine, or cabazitaxel; whereas, secondary transplants were treated with MTD of gemcitabine (daily or intermittent) or cytarabine (daily). In primary transplants, cytarabine did not significantly prolong survival compared with vehicle; whereas transplants treated with cabazitaxel (log-rank test, $P = 0.011$) achieved a significant

**Fig. 5** In vivo activity of gemcitabine and cabazitaxel. Kaplan–Meier analysis (top panels) of animal survival in **a** CHRF288-11-luciferase/YFP+ xenograft, **b** AMKL patient-derived xenograft, **c** ML-2-luciferase/YFP+ xenograft, **d** *CBFA2T3-GLIS2/JAK2*V617F-induced AMKL model, and **e**, **f** MLL-^PTD/wt^:FLT3^ITD/ITD^-double knock-in model. **a–c**, **e** Mice were randomized to receive vehicle (black), 50 mg kg^−1^ cytarabine (Ara-C, blue), gemcitabine 50 mg kg^−1^ (Gem, magenta), or cabazitaxel 5 mg kg^−1^ (Cab, gray) once every 4 days (q4d) for 3 weeks for up to 2 cycles; or **d**, **f** mice were randomized to receive vehicle (black), 50 mg kg^−1^ cytarabine (Ara-C, blue, solid line), 100 mg kg^−1^ cytarabine (Ara-C, blue, dashed line), or gemcitabine 3 mg kg^−1^ (Gem, magenta, solid line) once daily for 5 days for up to 3 cycles or 120 mg kg^−1^ gemcitabine (Gem, magenta, dashed line) once every 3 days (q3d) for 3 weeks for up to 1.5 cycles. All treatments were administered by intraperitoneal injection, black arrows indicate treatment schedule per model. Tumor burden was monitored (bottom panels) by **a**, **c** bioluminescent imaging, **b** detection of human CD45+ cells in peripheral blood, **d** detection of GFP/mCherry double positive (DP) cells in peripheral blood, or **e**, **f** detection of Ly5.2+ cells in peripheral blood. Student's *t* test; NS not significant; *$P < 0.05$; **$P < 0.01$; ***$P < 0.0001$

survival advantage (Fig. 5e, Supplementary Fig. 13). Gemcitabine provided the greatest survival advantage and significantly prolonged survival compared to all other treatment regimens (log-rank test; versus vehicle, $P = 0.0013$; versus cytarabine, $P = 0.024$; versus cabazitaxel, $P = 0.007$). In secondary transplants, cytarabine significantly prolonged survival compared with vehicle (log-rank test, $P = 0.009$); similarly daily MTD of gemcitabine significantly prolonged survival compared with vehicle (log-rank test, $P = 0.009$) (Fig. 5f). The intermittent MTD of gemcitabine was not well tolerated and mice only received 4 doses on this regimen (Supplementary Fig. 15); we attribute the lack of tolerability to the short time frame between treatment initiation and irradiation prior to transplant. Interestingly, however, weights fully recovered and mice had a profound response to the limited regimen achieving a significant survival advantage compared to vehicle (log-rank test, $P = 0.0006$) (Fig. 5f). All treatment groups had a significant reduction in tumor burden compared to vehicle on day 13 as indicated by decreased infiltration in the peripheral blood.

**JAK inhibitors are selective for AMKL**. In addition to identifying compounds with broad activity across high-risk subtypes, we also wanted to identify leads that were selective for specific subtypes or genetic abnormalities. Regarding the former, we found JAK inhibitors demonstrate potent selective activity for AMKL versus other subtypes and confirmed these findings (Supplementary Data 2, Supplementary Fig. 15). Due to the potential for clinical translation we prioritized evaluation of the two currently FDA-approved inhibitors (ruxolitinib, tofacitinib). Using an expanded panel of AMKL cells lines ($N = 9$), we found ruxolitinib to have more potent activity compared to tofacitinib (Fig. 6a). Interestingly, both compounds were not active in the two AMKL cell lines (Meg-01, MKPL) established from adult patients. We evaluated ruxolitinib sensitivity using primary blasts isolated from treatment naive CG/V617 tertiary transplants; we observed the primary murine blasts to be sensitive to ruxolitinib (Supplementary Fig. 16). These findings are consistent with the role JAK/STAT signaling plays in the underlying biology driving megakaryopoiesis and recurrent genetic aberrations in a significant portion of AMKL cases that result in upregulated JAK/STAT signaling[27,30]. To gain insight into the activity underlying the enhanced potency of ruxolitinib, we compared the binding affinity and kinase inhibition of JAK family members. In a binding assay, ruxolitinib has the highest affinity to the catalytic domain of JAK2 whereas tofacitinib demonstrated highest binding affinity for the catalytic domain of JAK3 (Table 2). In a kinase assay, ruxolitinib potently ($IC_{50} < 1$ nM) inhibited the enzymatic activity of JAK1, JAK2, and TYK2; in contrast tofacitinib potently inhibited JAK3 (Table 2). These findings are in agreement with the reported selectivity of both inhibitors[34,35].

Next, we compared the expression levels of JAK-STAT family members among a panel of AMKL (JAK mutated: CHRF288-11, CMK, CMY; CBFA2T3-GLIS2 positive: CMS, M07e, M-MOK, WSU) and non-AMKL (*MLL*r: ML-2; *MLL*r with FLT3-ITD:

MOLM-13, MV4-11; other: U937) cell lines using RNA-seq data to provide insight into the mechanism driving subtype selective activity. We found multiple JAK-STAT family members including JAK1 (2-fold), JAK3 (9.3-fold), STAT3 (3.5-fold), STAT5A (6-fold), STAT5B (2.9-fold) were significantly increased in AMKL compared to non-AMKL cell lines (Fig. 6b). We validated our findings using an expanded panel of AML cell lines and confirmed STAT5A expression is significantly higher at both the RNA and protein level in AMKL compared to non-AMKL, with the exception of HEL cells (positive control) that harbor a JAK2V617F mutation (Fig. 6c, e). Furthermore, we determined there is a high and significant correlation between STAT5A expression and ruxolitinib sensitivity in the AMKL cell lines (Fig. 6d); the sensitivity to ruxolitinib was also associated with a reduction in p-STAT5A signaling in AMKL regardless of JAK mutation status (Fig. 6e, f). Previous investigations have shown JAK signaling can be stimulated by a variety of cytokines/chemokines including BMP2[36], erythropoietin (EPO)[37], and thrombopoietin (TPO)[38,39]. Since TPO plays a primary role and EPO, to a lesser extent, in the maintenance of megakaryocytes, and given our previous report showing elevated BMP2 expression in *CBFA2T3-GLIS2*-positive AMKL patients[27] we determined if co-exposure would affect ruxolitinib sensitivity. We performed a cell viability assay using increasing concentrations of ruxolitinib with or without BMP2, EPO, or TPO. We found that TPO, a glycoprotein critical to megakaryocyte differentiation, caused a rightward shift of the dose-response curve resulting in a higher $IC_{50}$ concentration, an indication of resistance (Fig. 6g). The observed resistance correlated with an induction of p-STAT5A signaling following stimulation with TPO (Fig. 6h).

**Pharmacological assessment of ruxolitinib**. Next, we determined the pharmacokinetic properties of ruxolitinib following oral administration in the two mouse strains used for efficacy studies. A single dose of ruxolitinib was administered and total ruxolitinib concentrations in plasma were measured over a 2 h time-course. Concentration-time profiles and a summary of ruxolitinib pharmacokinetic parameters are shown in Fig. 7a, b and Supplementary Table 4. Although we observed strain differences, overall the area under the curve values and half-life approach those observed clinically in children with relapsed cancers[40].

**Ruxolitinib prolongs in vivo survival in multiple models**. Finally, we evaluated the in vivo efficacy of ruxolitinib using multiple murine models of AMKL. We found that ruxolitinib significantly prolonged survival compared to vehicle in all three models (log-rank test; CHRF288-11-Luc+, $P = 0.003$; CG-V617-Luc+ quaternary transplants, $P = 0.004$; AMKL PDX, $P = 0.005$) and significantly inhibited tumor burden compared to vehicle as indicated by a reduced bioluminescence and decreased infiltration in the peripheral blood (Fig. 7c–e).

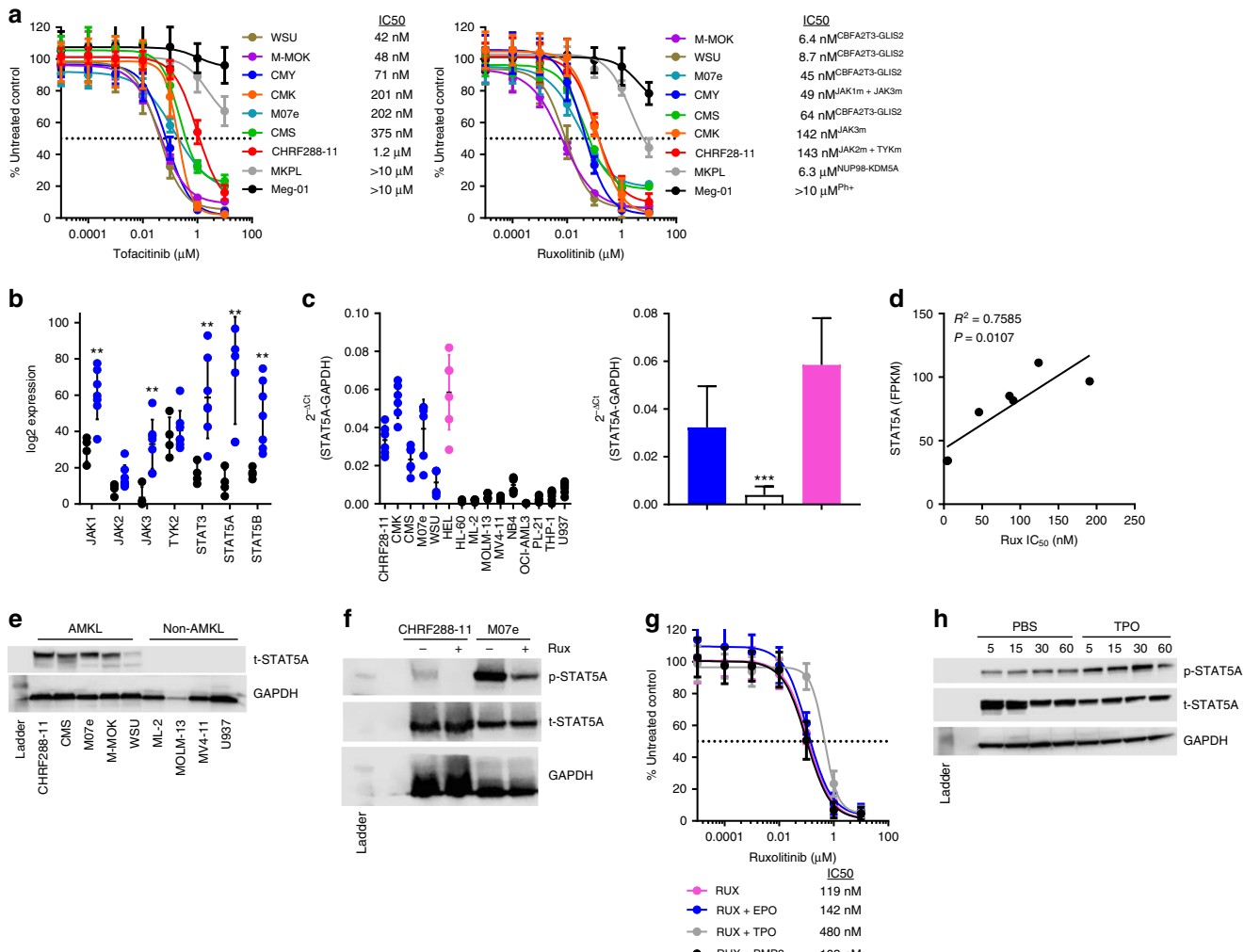

**Fig. 6** Activity of JAK inhibitors for AMKL. **a** AMKL cells were treated with increasing concentrations of vehicle or JAK inhibitor (tofacitinib, ruxolitinib) for 72 h, and cell viability was measured using Cell Titer Glo. Data are reported as percent control and represented as mean ± standard deviation (SD) of 3 independent experiments ($N = 18$ per concentration). The half maximal inhibitory concentration ($IC_{50}$, dotted line) was evaluated by nonlinear regression analysis using GraphPad Prism. **b** Expression of JAK-STAT family members was determined by RNAseq in AMKL cell lines (blue; CHRF288-11, CMK, CMS, CMY, M-MOK, M07e, WSU; $N = 7$) compared to non-AMKL cell lines (black; ML-2, MOLM-13, MV4-11, U937; $N = 4$). Data are mean ± standard deviation (SD). **c** Confirmation of *STAT5A* expression using a TaqMan expression assay was performed in an expanded panel of AML cell lines (blue circle, AMKL; magenta circle, HEL; black, non-AMKL) and normalized to *GAPDH* (left). Data are mean ± SD ($N = 6$). Mean per group ± SD (right). **d** Correlation analysis of *STAT5A* expression and ruxolitinib (Rux) $IC_{50}$ in AMKL cell lines was determined by Pearson correlation and linear regression. **e** Protein expression of total STAT5 (t-STAT5) in a panel of AMKL and non-AMKL cell lines. **f** AMKL cell lines (CHRF288-11, M07e) were exposed to their respective ruxolitinib $IC_{50}$ (143 and 45 nM) for 1 h then lysed. Western blot analysis was performed on whole cell lysate to evaluate protein expression of phospho-STAT5 (p-STAT5) and t-STAT5; GAPDH was used as a loading control. **g** CHRF288-11 cells were treated with increasing concentrations of ruxolitinib (magenta) with or without 100 ng mL$^{-1}$ of BMP2 (black), EPO (blue), or TPO (gray) for 72 h and cell viability was measured using Cell Titer Glo. Data are reported as percent control and represented as mean ± SD of 3 independent experiments ($N = 18$ per concentration). The $IC_{50}$ (dotted line) was evaluated by nonlinear regression analysis using GraphPad Prism. **h** CHRF288-11 cells were treated with 100 ng mL$^{-1}$ of TPO for up to 1 h and lysed; western blot analysis was performed on whole cell lysate to evaluate protein expression of p-STAT5 and t-STAT5; GAPDH was used as a loading control. Student's $t$ test; **$P < 0.01$; ***$P < 0.0001$

## Discussion

Despite many advances in the treatment of pediatric AML, the long-term survival is still unacceptably low and new therapeutic strategies are urgently needed for patients with high-risk subtypes. Using an integrated approach we have identified alternative chemotherapeutic regimens can be superior to standard of care and targeted agents may be useful for select subtypes. Collectively, our data suggest that treatment success could be improved through a repurposing strategy.

Nucleoside analogs have been widely used in the treatment of hematologic malignancies of which cytarabine is the mainstay of

therapy in AML[41]. Cytarabine and gemcitabine are structurally similar nucleoside analogs that require cellular uptake and activation through multiple intracellular phosphorylation steps. While both drugs are activated by the same enzymes, gemcitabine has additional mechanisms of action and a pattern of self-potentiation that is unique among nucleoside analogs[42,43]. Due to its higher affinity for deoxycytidine kinase gemcitabine undergoes greater activation to mono- and di-phosphorylated metabolites compared to cytarabine and fludarabine[43,44]. Further, dFdCDP is reported to inhibit ribonucleotide reductase, an enzyme in the nucleotide pathway and critical for management of

**Table 2 Ruxolitinib and tofacitinib in vitro kinase binding and inhibition**

| Target in binding assay[a] | $K_d$ Ruxolitinib (nM) | $K_d$ Tofacitinib (nM) | Target in kinase assay | IC$_{50}$ Ruxolitinib (nM) | IC$_{50}$ Tofacitinib (nM) |
|---|---|---|---|---|---|
| JAK1 H1 domain-catalytic | 11 | 5.2 | JAK1 | 0.54 | 1.9 |
| JAK1 JH2 domain-pseudokinase | 15,000 | 30,000 | JAK2 | 0.47 | 6.2 |
| JAK2 JH1 domain-catalytic | 0.054 | 0.59 | JAK2V617F | 1.5 | 2.4 |
| JAK3 JH1 domain-catalytic | 0.87 | 0.18 | JAK3 | 14.7 | 0.97 |
| TYK2 JH1 domain-catalytic | 0.25 | 6.4 | TYK2 | 0.45 | 25 |
| TYK2 JH2 domain-pseudokinase | 2200 | 30,000 | | | |

[a]Data obtained from KdELECT

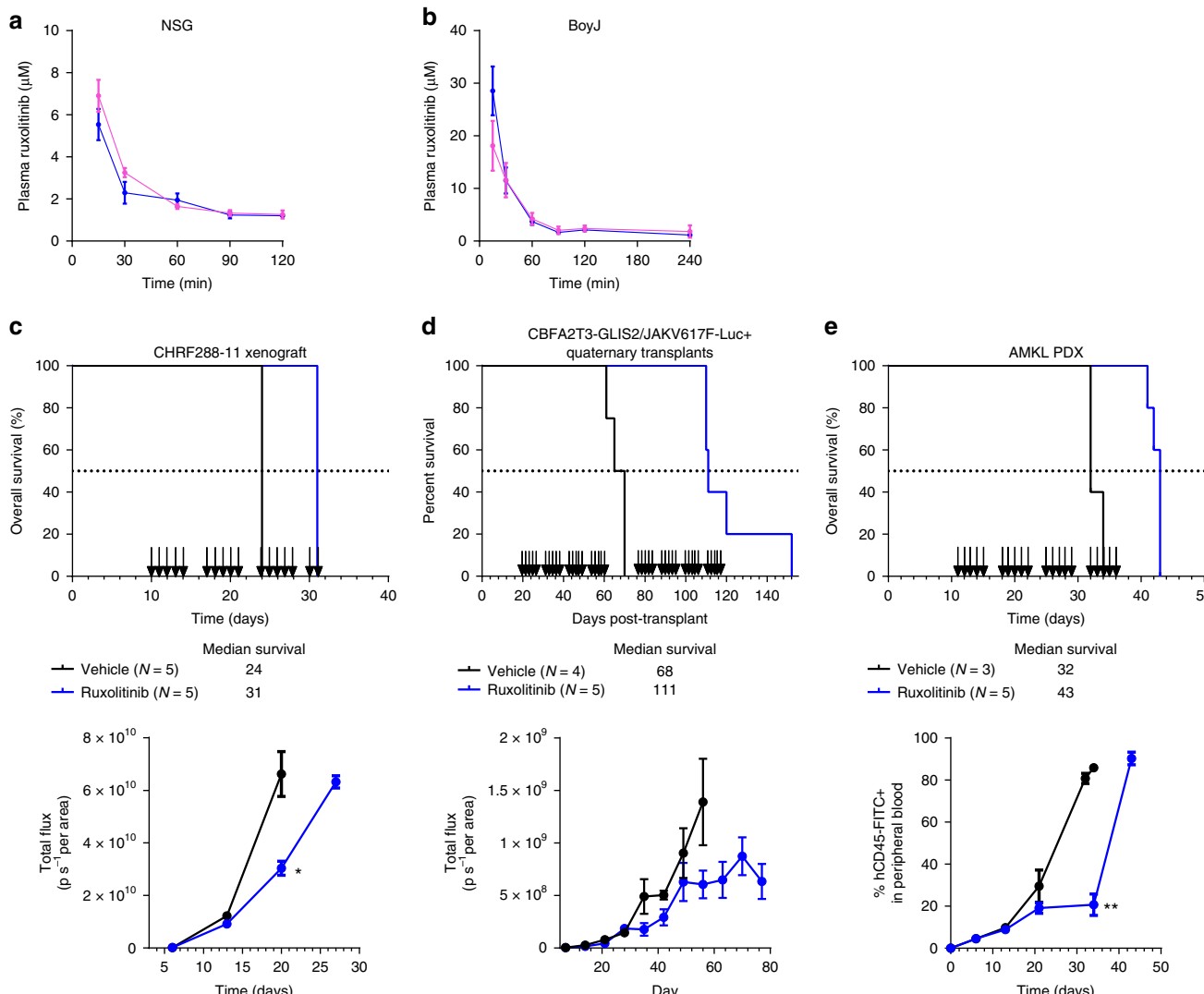

**Fig. 7** Pharmacokinetics and in vivo activity of ruxolitinib. Pharmacokinetic profile in female (magenta) and male (blue) **a** NSG and **b** BoyJ mice treated with a single dose of ruxolitinib (60 mg kg$^{-1}$, oral gavage). Serial blood sampling was performed and ruxolitinib plasma concentrations were determined by LC-MS/MS. Data are mean ± standard error (SE) (N = 5 per gender). Concentration-time data were analyzed by non-compartmental analysis and pharmacokinetic parameters were calculated in WinNonLin. **c–e**, top Kaplan–Meier analysis of animal survival in **c** CHRF288–11-luciferase/YFP+ xenograft, **d** *CBFA2T3-GLIS2/JAK2*V617F-induced AMKL quaternary transplants, and **e** AMKL patient-derived xenograft. Mice were randomized to receive vehicle (black) or ruxolitinib (blue; 60 mg kg$^{-1}$) twice daily for 5 days for up to 4 weeks; black arrows indicate treatment schedule per model. **c–e**, bottom Tumor burden was monitored by **c**, **d** bioluminescent imaging or **e** detection of human CD45+ cells in peripheral blood; data are mean ± SE. Student's *t* test; *P < 0.01; **P < 0.0001

deoxynucleotide pools in cancer cells[45]. Incorporation of dFdCTP into DNA results in masked chain termination, where one additional deoxynucleotide is incorporated before termination of DNA synthesis. This specific type of nucleotide linkage masks the gemcitabine nucleotide and prevents recognition by exonuclease making repair difficult[44,46]. These mechanisms are likely contributing to the significantly greater anti-leukemic activity of gemcitabine that we observed.

Besides inherent mechanisms of action we attribute the enhanced activity of gemcitabine to mechanisms of drug uptake and efflux, processes that contribute to intracellular accumulation. This is supported by our observations demonstrating a significantly greater accumulation of gemcitabine and metabolites. Our observation that the rate of accumulation of dFdCTP was linear up to 2 h is consistent with the results of Gandhi and Plunkett, who found accumulation of gemcitabine to be linear up to 3 h in K562 cells[47]. Together the rate of drug uptake, efficiency of phosphorylation, and efflux influence the rate of triphosphate formation and cellular retention for both cytarabine and gemcitabine which ultimately impacts cytotoxicity. We performed uptake assays at increased concentrations of cytarabine and demonstrated that a 10-fold higher was necessary to achieve similar intracellular exposure compared to gemcitabine. These results are consistent with a previous report by Hertel et al.[48], which demonstrated a minimum effective concentration of cytarabine is 10-fold higher than gemcitabine in CCRF-CEM cells (T lymphoblastoid cell line) and our own data showing a higher $IC_{50}$ for cytarabine compared to gemcitabine. Here, we show the transport efficiency of both drugs and demonstrate that even at escalated concentrations of cytarabine, AML cells have the capacity to transport gemcitabine more efficiently which contributes to the observed differences in intracellular accumulation and cytotoxicity.

While the human ENT1 is thought to be the primary transporter mediating drug influx, we recently identified that entry of cytarabine and several structurally related nucleosides, including gemcitabine, is facilitated by the ergothioneine uptake transporter OCTN1 (SLC22A4; ETT); and low expression of OCTN1 in leukemia cells is a strong predictor of poor survival in multiple cohorts of patients with AML treated with cytarabine-based regimens[20]. In a parallel study, we reported that cytarabine and Ara-CMP are sensitive to multi-drug resistant protein 4 (MRP4)-mediated efflux, thereby decreasing its cytotoxic response against AML blasts[49]. In contrast, a similar study found that gemcitabine and its metabolites are not effluxed by MRP4 or MRP5[50]. It can thus be postulated that differential expression and activity of uptake transporters and their affinities for nucleoside analogs, phosphorylating enzymes that result in active metabolites, and MRPs that mediate drug efflux, play a crucial role in the differential anti-leukemic activity of these compounds. This is consistent with our previous report showing over-expression of OCTN1 in HEK293 cells resulted in increased sensitivity to multiple nucleoside analogs, including cytarabine[20] and our current findings that inhibition of OCTN1 and ENT1 reduced the accumulation of AML cells in S-phase with gemcitabine or cytarabine treatment.

A phase II study of gemcitabine as a single agent demonstrated no significant activity in relapsed and refractory childhood acute leukemias[51]. However, these results were not anticipated and the lack of activity was unclear though patients were heavily pre-treated and gemcitabine was administered as a continuous 360 min infusion rather than a more contemporary intermittent schedule with a 30 min to 1 h infusion.

Our preclinical studies suggest that cabazitaxel triggers a mitotic cell cycle arrest which leads to cell death and can significantly prolong survival in multiple murine models of AML.

Though pre-clinical studies have shown that paclitaxel and docetaxel were efficient in killing pediatric solid tumors and acute leukemia, tubulin-stabilizing agents have undergone limited clinical testing in pediatric oncology[52,53]. There is evidence for taxanes exhibiting synergistic toxicity when combined with other cytotoxic agents such as cisplatin in highly resistant brain tumors[52]. It has been suggested that limited success with taxanes may be attributed, in part, to early phase trials conducted in heavily pre-treated patients[52,54]. More recently, results from the NCT01751308 clinical trial evaluated the safety and efficacy of cabazitaxel in pediatric patients with refractory solid tumors. Although no objective responses were observed in the phase I dose-escalation portion of the trial, a MTD was established in this pediatric population. The lack of neurotoxicity in this clinical trial is of potential relevance in regard to safety concerns of taxanes in children and allowing for potential clinical translation in a high-risk and/or relapsed/refractory setting.

In this study, we also observed multiple kinase inhibitors targeting a variety of kinases involved in mitosis including alisertib (aurora A; active 4/8), barasertib (aurora B; active 3/8), MK-1775 (Wee1; active 7/8) rigosertib (polo-like kinase; active 7/8), and volasertib (polo-like kinase; active 7/8) and motor protein inhibitor ARRY-520 (Eg5; active 3/8) to have anti-leukemic activity, highlighting a vulnerability inherent to AML, among other cancers (Supplementary Data 2). Several of these compounds are in late stage clinical development, specifically polo-like kinase 1 inhibitors, which demonstrated broad activity in both our HTS and in primary patient samples and have been recognized as Innovative Therapy in Leukemia by the FDA; volasertib has been designated an orphan drug and received breakthrough therapy status for the treatment of AML by the FDA[55,56]. However, a primary analysis of a phase III study for the treatment of AML with volasertib plus low-dose cytarabine (LDAC) versus placebo plus LDAC did not meet the primary endpoint and patients receiving volasertib plus LDAC were at higher risk for fatal infections[57]. Therefore, further work will be required to refine our understanding of cell cycle regulation and aberrant function of key regulators that may have a broad role in leukemogenesis or that may associate with specific subtypes and may guide our selection of the optimal mitotic-targeted agent for the treatment of AML.

The clinically used JAK inhibitor ruxolitinib demonstrated selective activity for AMKL and had slightly greater activity compared to tofacitinib. Interestingly, we found a significantly higher expression of STAT5A in AMKL compared to non-AMKL cell lines, including those with FLT3-ITD; and expression highly correlated with ruxolitinib sensitivity compared to other JAK-STAT family members. Our findings suggest that targeting STAT5A is integral to the underlying mechanism driving ruxolitinib's selective activity in AMKL. While ruxolitinib has been approved for intermediate to high-risk myelofibrosis, the Children's Oncology Group has completed a phase 1 study in relapse/refractory solid tumors, leukemia, and myeloproliferative neoplasms where a tolerable pediatric dose and schedule was established[40]; current clinical trials are evaluating ruxolitinib in combination with chemotherapy for the treatment of pediatric lymphoblastic leukemia (NTC02723994).

Collectively, these data provide a rationale for the evaluation of gemcitabine and cabazitaxel in pediatric AML. These agents demonstrated broad activity across multiple high-risk subtypes and may exhibit therapeutic benefit in other subtypes of childhood AML. Similarly, our data provide justification for the clinical evaluation of ruxolitinib for the treatment of pediatric AMKL. Further evaluation regarding optimal dose and schedule in addition to the identification of drug combinations will inform future clinical trial design.

## Methods

**Cell culture and reagents.** Human AML cell lines HEL 92.1.7 (HEL; ATCC TIB-180), HL-60 (ATCC CCL-240), Meg-01 (ATCC CRL-2021), MV4-11 (ATCC CRL-9591), THP-1 (ATCC TIB-202), and U937 (ATCC CRL-1593.2) cells were obtained from ATCC (Manassas, VA); ML-2 (ACC-15), M07e (ACC-104), MOLM-13 (ACC-554), NB4 (ACC-207), and PL-21 (ACC-536) cells were obtained from DSMZ (Braunschweig, Germany); MKPL-1 (MKPL; JCRB1325) cell were obtained from Japanese Collection of Research Bioresources Cell Bank (Osaka, Japan); M-MOK (RCB2534) cells were obtained from RIKEN BioResource Research Center (Tsukuba, Japan); WSU-AML cells were obtained from Asterand Bioscience (Hicksville, NY); CHRF288-11 cells were a kind gift from Dr. Tanja Gruber (St. Jude Children's Research Hospital [SJCRH], Memphis, TN); CMK, CMS, and CMY cells were a kind gift from Dr. Jeffrey Taub (Karmanos Cancer Center Institute, Detroit, MI); OCI-AML3 were a kind gift from Dr. Brian Sorrentino (SJCRH). Note, the U937 cell line has been listed in the database of commonly misidentified cell lines; this cell line has been used in this report as a representation of a more mature pro-monocytic leukemia, authentication was confirmed prior to use. All cell lines are tested regularly for mycoplasma contamination using a commercially available kit (Lonza, Walkersville, MD) and verified by STR profile. Passages are kept to a minimum and cells are not used beyond passage 30. Cells were cultured in RPMI1640 media with L-glutamine (Life Technologies, Grand Island, NY) supplemented with 10% fetal bovine serum (FBS) (Atlanta Biologicals, Lawrenceville, GA) at 37 °C in a humidified incubator with 5% $CO_2$. M07e cells were supplemented with 10 ng/mL IL-3 (Life Technologies); M-MOK cells were supplemented with 10 ng/mL GM-CSF (Life Technologies); MKPL cells were supplemented with 20% FBS. Primary AML blasts from 10 patients with $FLT3$-ITD ($N = 3$), $MLLr$ leukemia (t(6;11); $N = 6$), or AMKL ($N = 2$) were obtained with patient or parent/guardian-provided informed consent under protocols approved by the Institutional Review Board at SJCRH. Mesenchymal stromal cells (MSC) were obtained from Dr. Dario Campana and cultured in RPMI1640 with L-glutamine and supplemented with 10% FBS and 1 μM hydrocortisone (complete MSC media) (Sigma-Aldrich, St. Louis, MO).

For low-throughput in vitro studies, artesunate, cytarabine, and docetaxel were purchased from Sigma-Aldrich; gemcitabine, cabazitaxel, paclitaxel, panobinostat, vorinostat, ABT-199, bortezomib, carfilzomib, GDC-0941, ibrutinib, trametinib, and sorafenib were purchased from LC Labs (Woburn, MA); romidepsin, RG7112, BMN673, MK-1775, volasertib, alisertib, palbociclib, clofarabine, and fludarabine were purchased from Selleck Chemicals (Houston, TX). All compounds were prepared in DMSO, except nucleoside analogs which were prepared in phosphate buffered saline (PBS).

**High-throughput screen.** Prior to screening, the optimal growth conditions including plating density and DMSO-sensitivity for all cell lines were determined. Briefly, for primary screening, cells were seeded in 25 μL culture medium in each well of 384-well plates (Corning, Tewksbury, MA) using an automated plate filler (Wellmate, ThermoFisher Scientific, Waltham, MA). After 24 h, 25 nL of solution containing a panel of compounds ($N = 7389$) were pin transferred into 384-well plates resulting in approximately 10 μM final drug concentration. Each plate included DMSO only negative controls and cycloheximide single point (0.5 μM) and dose-response (0.01 nM to 0.5 μM) positive controls. Cell viability was determined using Cell Titer Glo (CTG; Promega, Madison, WI) and an automated Envision plate reader (Perkin-Elmer, Waltham, MA) after a 72-h incubation. Luminescence data were normalized by $\log_{10}$ transformation and the percentage inhibition = 100 × (sample result − negative control mean)/(positive control mean − negative control mean) calculated. Similarly, secondary screens were conducted in a dose-response manner (10-point curve, 1 nM to 10 μM); a limited number of compounds ($N = 458$) were applied in serial dilution (0.5 nM to 10 μM final concentration) and repeated in triplicate. Compounds selection for the secondary screen was based on the following: (1) demonstration of >50% inhibition in more than one cell line in the primary screen; (2) currently or previously evaluated in clinical phase testing; (3) analogs of compounds in the primary screen showing activity that were not included; (4) compound classes of interest not included in primary screen. For dose-response drugging, each compound is drugged against multiple wells per plate at different concentrations per well. After a 72-h incubation, cell viability was determined by the CTG/EnVision system. Dose-response curves were fit using the *drc* package in R. The four parameters sigmoidal function *LL2.4* was used with the following constraints: −10 ≤ hill slope ≤ 0; 0 ≤ y0 ≤ max median normalized % activity; 0 ≤ yFin ≤ max median normalized % activity; $10^{10} ≤ ec_{50} ≤ 10^{-4}$ (which equates to the range of drug concentrations tested in this experiment). Values from the highest concentrations tested for each compound are weighted at 10% to reduce curve fitting artifacts. In the event of a failure to fit a sigmoidal dose-response curve, the *smooth.spline* option in R was used to fit a curve that could be used to determine the area under the curve.

We used the following criteria for the advancement of compounds from the secondary screen that demonstrated broad activity across subtypes (e.g., EC50 < 1 μM in all cell lines, listed in Table 1): (1) the drug is FDA-approved, (2) a pediatric dose has been determined or the agent is in phase 1 pediatric testing, and (3) the drug is not currently used clinically or under investigation for the treatment of adult or pediatric AML.

**Cell viability assays.** Cell viability was evaluated using 3-(4,5-dimethylthiazol-2-yl)-2,5-diphenyltetrazolium bromide (MTT) reagent (Sigma-Aldrich) or Cell Titer Glo (Promega) in a low-throughput manner. Cell lines were seeded in 96-well plates and treated with increasing concentrations of drug for 72 h. Three independent experiments were performed (18 total replicates per concentration). At 68 h after treatment, 10 μL of MTT reagent (5 mg/mL MTT in PBS) was added to each well and plates were incubated at 37 °C with 5% $CO_2$ for 4 h; formazan crystals were solubilized with 100 μL of acidified isopropanol (Sigma-Aldrich). Cell Titer Glo was used according to the manufacturer's instructions. The absorbance or luminescence was measured using a Synergy H4 (Biotek Instruments, Inc., Winooski, VT). The half maximal inhibitory concentration ($IC_{50}$) was evaluated by nonlinear regression analysis in the software program GraphPad Prism version 5.04 (GraphPad Software, La Jolla, CA).

**Ex vivo drug treatments with primary AML blast samples.** Bone marrow MSCs were plated in 96-well plates at a density of 10,000 cells/well in complete MSC media. After 48 h, complete MSC media was removed and replaced by RPMI1640 with L-glutamine supplemented with 10% FBS and 1% penicillin/streptomycin (ThermoFisher Scientific) overnight. Primary AML blast cells were seeded at a density of 100,000 cells/well and treated with increasing concentrations of single agent (6 concentration points; 10-fold dilution). After 96 h treatment, primary AML blasts were separated from the MSC by manual pipetting. Cell Titer Glo was used according to the manufacturer's instructions and luminescence was measured. $IC_{50}$ concentrations were calculated, as described above. Plotly (Plotly, Inc., Montreal, Quebec) was used to generate a heatmap.

**Cellular uptake and accumulation studies.** Logarithmically growing cells (3 × $10^6$), were washed with PBS and seeded in 12-well plates with serum-free medium containing a mixture of unlabeled and radioactively labeled cytarabine (total concentration 1, 10, or 100 μM) or gemcitabine (total concentration 1 μM) and were incubated for 5–120 min at 37 °C with 5% $CO_2$. $^3$H-cytarabine (15 $C_i$ mmol$^{-1}$) and $^3$H-gemcitabine (16.2 $C_i$ mmol$^{-1}$) were purchased from Moravek Biochemicals (Brea, CA, USA). Cells were washed twice with ice-cold PBS; cell pellets were solubilized in 400 μL 1 N NaOH and agitated (300 rpm) at room temperature for 2 h. Then samples were neutralized with 200 μL 1 M HCl; a 25 μL aliquot of lysate was used to estimate protein concentration using a Pierce BCA protein assay kit (Thermo Scientific). Total radioactivity was measured by a Tri-Carb 4810TR liquid scintillation counter (Perkin-Elmer) after mixing the sample with 4 mL of Emulsifier Safe (Perkin Elmer); the results were normalized to total protein content as measured by a Pierce BCA protein assay kit. Three independent experiments were performed in triplicate.

Total radioactivity was also detected in the nuclear cell fraction, for these experiments 5 × $10^6$ cells were treated with a mixture of unlabeled and radioactively labeled cytarabine or gemcitabine (total concentration 1 μM) for 2 h, washed twice with ice-cold PBS then resuspended in 500 μL 1× hypotonic buffer (20 mM Tris–HCl, pH 7.4, 10 mM NaCl, 3 mM $MgCl_2$) and incubated for 15 min on ice. Next, 25 μL of detergent (10% $NP_4O$, Thermo Scientific) was added, then samples were vortexed for 10 s at highest setting followed by centrifugation for 10 min at 3000 rpm at 4 °C. The nuclear pellet was resuspended in 50 μL complete extract ion buffer (cell extraction buffer [Life Technologies], protease inhibitor cocktail [Calbiocehem], 1 mM PMSF) and incubated for 30 min on ice with vortexing at 10 min intervals then centrifuged for 30 min at 14,000×g at 4 °C. The supernatant containing the nuclear fraction was transferred to a clean tube; 40 μL was used to detect radioactivity using the liquid scintillation counter. A Pierce BCA protein assay was performed on the cytoplasmic fraction of each sample and results were normalized to total protein content. Three independent experiments were performed in triplicate.

**HPLC to detect compounds and metabolites.** Logarithmically growing cells (3 × $10^6$), were washed and seeded in 12-well plates with serum-free medium containing a mixture of unlabeled and radioactively labeled cytarabine or gemcitabine (total concentration 1 μM) and were incubated for 2 h at 37 °C with 5% $CO_2$. Cells were washed twice in ice-cold PBS then resuspended in 300 μL of buffer containing 70% methanol and 30% 15 mM Tris (pH 7.4) and shaken (300 rpm) for 10 min at 4 °C. After centrifugation, 50 μL of cell extract was removed for protein concentration measurement using a Pierce BCA protein assay kit. Intracellular cytarabine, gemcitabine, and phosphorylated metabolites were measured by HPLC. The following standards were used: Ara-C, cytidine 5′-monophosphate disodium salt (Sigma), cytidine 5′-diphosphocholine sodium salt (Sigma), cytidine 5′-diphosphate trisodium salt (Sigma), cytidine 5′-triphosphate disodium salt (Sigma), gemcitabine hydrochloride (Sigma), gemcitabine monophosphate (Toronto Research Chemicals), gemcitabine diphosphate trimethylamine salt (Toronto Research Chemicals), gemcitabine triphosphate ditriethylamine (Toronto Research Chemicals). All standards were prepared to a concentration of 1 mg mL$^{-1}$ in water. Analysis was carried out with a Hewlett Packard Series 1100 HPLC. Separation was achieved on a PARTISIL SAX anion exchange HPLC column (Whatman 4.6 mm × 250 mm × 10 μM particles) at room temperature. Mobile phase A is 0.5 mM $NH_4H_2PO_4$, and mobile phase B is 500 mM $NH_4H_2PO_4$ pH 3.4. Flow rate is varied from 0.5 to 1.0 mL min$^{-1}$ at a temperature of 40 °C. The gradient was modified in

such a manner that it is held isocratically from 0 to 10 min at a flow rate of 0.5 mL min$^{-1}$. At 10 min the flow changes to 1.0 mL min$^{-1}$ and a linear gradient runs from 0 to 100% B from 10.01 to 40 min. Fractions were collected directly into scintillation vials at 1 min intervals using an Eldex Universal Fraction Collector. The total run time was 70 min. After addition of Scintisafe 30% scintillation fluid (Perkin Elmer), samples were vortexed and radioactivity was measured on a Tri-Carb 4810TR liquid scintillation counter. Total radioactivity was expressed as disintegrations per minute (DPMs) and normalized to protein concentration. Three independent experiments were performed.

**Silencing of ENT1 and OCTN1 expression.** The on-target plus SMART pool human SLC22A4 siRNA for OCTN1 silencing (Dharmacon, Inc./Horizon, Cambridge, UK), and the Mission siRNAs targeting ENT1 or negative non-targeting control (Sigma-Aldrich) were used in all experiments. MOLM-13 cells were transfected with siRNA control, siRNA OCTN1 or siRNA ENT1 using Nucleofector II and Nucleofector Kit C (Lonza) program X-001, according to the manufacturer's protocol. Briefly, MOLM-13 cells ($3 \times 10^6$) were transfected with 200–500 nM siRNA for each sample; suppression of OCTN1 (Hs00268200_m1) and ENT1 (Hs01085704_g1) was evaluated using a TaqMan qRT-PCR assay at 48 h post-transfection; expression levels were normalized to GAPDH (Hs02758991_g1; VIC). Cytotoxicity studies were initiated in MOLM-13 cells 24 h after transfection. Asynchronous cells were treated with PBS, 10 nM gemcitabine, or 100 nM cytarabine for 16 h; then cells were harvested for cell cycle distribution as described below.

**Cell cycle distribution and induction of mitosis.** Asynchronous cells were treated with 5 or 50 nM cabazitaxel for up to 24 h; DMSO was the control agent. At indicated time points (4, 8, 12, 16, 24 h) cells were collected and washed once with 0.1% EDTA (Ricca Chemical Company, Arlington, TX) in PBS then washed once with PBS only. Next, cells were fixed with ice-cold 70% ethanol for 30 min on ice or stored at −20 °C for up to 1 month. Cells were spun down at $450 \times g$ for 10 min and stained with DAPI (DAPI dilactate concentration 1 μg mL$^{-1}$ [ThermoFisher Scientific]) in 0.1% Triton-X [ThermoFisher Scientific]/PBS for 30 min at room temperature and protected from light. The DNA content was determined using a BD LSR II flow cytometer (BD Biosciences, San Jose, CA, USA). The cell cycle distribution was analyzed using FlowJo v10.0.08 software (FlowJo, LLC, Ashland, OR). For biparametric analysis of mitosis, cells were fixed with 16% paraformaldehyde (Avantor, Center Valley, PA) in PBS at room temperature for 10 min, washed twice with PBS then permeabilized with ice-cold 95% methanol (ThermoFisher Scientific) and incubated at −20 °C. Cells were rehydrated in FACS buffer (PBS + 4% FBS) and stained with cyclin B1-APC (Cell Signaling Technology, Inc., Danvers, MA) and phospho-histone H3-FITC (Cell Signaling Technology, Inc.). Two-color flow cytometry was performed using a BD LSR II flow cytometer (BD Biosciences) and the data was analyzed using FlowJo (FlowJo, LLC).

**Binding assay.** Binding of ruxolitinib and tofacitinib to purified JAK1, JAK2, JAK3, and TYK2 kinases were performed using a commercially available KdELECT assay (DiscoverRx, Fremont, CA), as previously described[58]. The binding constant ($K_d$) was calculated with a standard dose-response curve using the Hill equation (slope set to −1); curves were fitted using a non-linear least square fit with the Levenberg–Marquardt algorithm.

**Kinase assay.** In vitro profiling of JAK1, JAK2, JAK2V617F, JAK3, and TYK2 kinase were performed at Reaction Biology Corporation[59]. Briefly, specific kinase/substrate pairs were prepared in fresh base reaction buffer (20 mM HEPES at pH 7.5, 10 mM MgCl$_2$, 1 mM EGTA, 0.02% Brij35, 0.02 mg/mL BSA, 0.1 mM Na$_3$VO$_4$, 2 mM DTT, 1% DMSO). Compounds were delivered into the reaction and incubated for 20 min. Next, $^{33}$P-ATP was delivered to the reaction mixture to initiate the reaction and incubated for 2 h at room temperature. Reactions were spotted onto P81 ion exchange paper and kinase activity was detected by filter-binding method.

**RNA isolation and qRT-PCR.** RNA was isolated from cell lines using Trizol (Invitrogen) chloroform (Sigma-Aldrich) extraction. cDNA was generated from 1 μg of RNA using the SuperScript IV First-Strand Synthesis System (Invitrogen) according to the manufacturer's instructions. PCR with reverse transcription was performed using TaqMan master mix (Life Technologies) at 2× dilution; TaqMan expression assays (Applied Biosystems) for OCTN1 (Hs00268200_m1; FAM), ENT1 (Hs01085704_g1; FAM), STA5A (Hs00559643_m1; FAM) and GAPDH (Hs02758991_g1; VIC) were used. A QuantStudio 3 system (Life Technologies) was used to perform quantitative PCR. Analyses were performed in technical triplicates using the delta Ct method.

**RNA-seq library preparation.** Gene expression profiles for JAK-STAT family members were done utilizing RNA-seq data. RNA sequencing libraries were prepared using 1 μg of RNA using TruSeq RNA Prep v2 kits according to the manufacturer's instructions (Illumina, Inc., San Diego, CA). Paired-end sequencing was performed on an Illumina HiSeq2000 using TruSeq SBS v3 reagents kits according

to the manufacturer's instructions, at the SJCRH Hartwell Center. To quantify the expression level of each gene, we performed the following procedures: (1) for each annotated exon in the RefSeq genes, we obtained the average coverage; (2) due to the high variations of coverage across different exons for each gene, we used the average coverage of the best covered exon as the expression level for each gene. After we combine the expression for all 12 samples, we only retained those genes whose expression level is at least 10 in at least one sample in order to exclude genes that are unexpressed or poorly expression across all samples. For gene expression comparisons, we obtained counts of the number of reads per gene and carried out fragment per kilobase mapped (FPKM) normalization and a quantile normalization was performed to adjust different sequencing depths for each sample.

**Murine bone marrow transduction and transplantation.** All animal studies were performed in accordance with Animal Care and Use Programs under protocols approved by the Institutional Animal Care and Use Committee at SJCRH or The Ohio State University (OSU). We have complied with the relevant ethical considerations for animal research overseen by this committee. Bone marrow from 4–6-week-old female C57BL/6 mice was harvested, lineage depleted (Lineage Cell Depletion Kit, Miltenyi Biotec Inc., Auburn, CA) and cultured in the presence of cytokines for 24 h prior to transduction on RetroNectin (Takara Bio Inc., Kyoto, Japan). Ectopic envelope-pseudotyped retroviral vectors were produced and replication-incompetent supernatant was made by transiently transfecting 293 T cells[29,60]. For primary bone marrow transplants, $0.1 \times 10^6$ single (GFP, mCherry) or double-positive cells with $1 \times 10^6$ protected cells were injected intravenously via tail vein on day 0. Recipient mice were conditioned with total body irradiation at a dose of 1100 rad divided on days −1 and 0. For secondary-quaternary transplants, $2 \times 10^6$ bone marrow and splenic cells from moribund mice were isolated and injected intravenously into recipient mice conditioned with 500 rad. Alternatively, double-positive cells isolated from secondary transplants and were transduced with a luciferase-BFP to generate CG/V617-Luc+ AMKL blasts; GFP/mCherry/BFP triple-positive cells were purified by flow cytometry then injected into conditioned tertiary or quaternary recipients.

**Establishment of in vivo gemcitabine maximum tolerated dose.** All animal studies were performed in accordance with Animal Care and Use Programs under protocols approved by the Institutional Animal Care and Use Committee at OSU. We have complied with the relevant ethical considerations for animal research overseen by this committee. Mice were housed in a barrier facility. Non-tumor bearing Ly5.2 C57Bl/6 (BoyJ) mice were treated with 100, 120, or 140 mg kg$^{-1}$ gemcitabine on an intermittent schedule every 3 days for 3 weeks; 3 or 6 mg kg$^{-1}$ doses were used to evaluate daily for 5 days schedule. All doses were administered by intraperitoneal injection; tolerability was assessed by daily weights and observation of changes in body condition.

**In vivo efficacy studies in murine models of AML.** All animal studies were performed in accordance with Animal Care and Use Programs under protocols approved by the Institutional Animal Care and Use Committee at OSU. We have complied with the relevant ethical considerations for animal research overseen by this committee. Mice were housed in a barrier facility. For cell line xenografts, NSG mice were procured from the OSU Comprehensive Cancer Center Target Validation Shared Resource. To label CHRF288-11 cells, lentivirus encoding firefly luciferase (LUC) and YFP (CL20SF2-Luc2AYFP with VSV-G envelope; SJCRH Vector Core) was used. Briefly, CHRF288-11 cells were transduced by spinfection using retronectin coated plates and 10 μL of concentrated viral supernatant obtained from the SJCRH Vector Core. Following transduction, cells were single cell sorted into each well of a 96-well plate by flow cytometry, individual clones were expanded, then evaluated in vivo for engraftment. For efficacy studies, mice were injected intravenously via tail vein with $5 \times 10^6$ CHRF288-11-LUC/YFP+ or ML-2-LUC/YFP+ cells; engraftment was monitored by noninvasive imaging performed once weekly starting on day 6 or 7 after injection. The Luciferase substrate D-luciferin firefly potassium salt (Gold Biotechnology, Inc., St. Louis, MO, USA) was administered by intraperitoneal injection at a dose of 150 mg/kg. Mice were anesthetized by 1.5–2.5% isoflurane (Baxter, Deerfield, IL, USA) inhalation; bioluminescence was determined 5 min later using a Xenogen IVIS-200 imaging system (Perkin Elmer). Total body bioluminescence was quantified for the body area that included each mouse in its entirety (Living Image 4.3.1, Perkin Elmer). For AMKL PDX, $1 \times 10^6$ cells whole bone marrow and spleen cells isolated from an individual primary transplant recipient were injected intravenously into sublethally irradiated (200 rad) NSG-SGM3 female mice that were 8–12 weeks old for expansion. For efficacy studies, $1 \times 10^6$ cells whole bone marrow and spleen cells isolated from an individual secondary transplant recipient were injected intravenously into sublethally irradiated (200 rad) NSG-SGM3 female mice that were 8–12 weeks old. Transplant studies used whole bone marrow and spleen cells from an individual Mll$^{PTD/wt}$:Flt3$^{ITD/ITD}$ double knock-in (dKI) mouse or individual CG-V617F-Luc+ tertiary transplant recipient that had been stored in liquid nitrogen. After thawing, $2 \times 10^6$ cells were injected intravenously into sublethally irradiated (500 rad) Ly5.1+ syngeneic female mice that were 8–12 weeks old. Due to the severely enlarged spleens (weight >1 g) in CG-V617F-Luc transplant recipients imaging is discontinued after day 45 and tumor burden is monitored by

peripheral blood for the remainder of the study. To establish the PDX model, a single vial of patient cells was thawed and engrafted in primary recipients then further expanded in secondary recipients; whole bone marrow and spleen cells were stored in liquid nitrogen. On day −1, 8–12-week-old triple transgenic NSG-SGM3 mice (The Jackson Laboratory; Bar Harbor, ME, USA) expressing human IL-3, GM-CSF, and SCF were sublethally irradiated (200 rad) with one dose of total body irradiation. On day 0, $2 \times 10^6$ cells were injected intravenously via tail vein into tertiary recipients. Leukemia progression was monitored by white blood cell (WBC) count analysis and/or detection of Ly5.2+ cells, GFP/mCherry-double positive, or human CD45 positive cells in peripheral blood by flow cytometry weekly. All mice were observed daily and humanely euthanized when showing signs of progressive disease including, hind limb paralysis, weight loss more than 15%, and lethargy.

To evaluate anti-leukemic activity NSG/NSG-SGM3 mice and dKI AML primary recipients were randomly assigned upon detection of significant tumor burden, based on imaging data or peripheral blood analysis, to receive vehicle, cytarabine or gemcitabine 50 mg kg$^{-1}$, or cabazitaxel 5 mg kg$^{-1}$ once every 4 days for 3 weeks; some mice were administered two cycles of dosing regimen with 1 week off between cycles as indicated. CG-V617-Luc+ quaternary recipients and dKI secondary recipients were administered 50 or 100 mg kg$^{-1}$ cytarabine once daily for 5 days, 3 mg kg$^{-1}$ gemcitabine once daily for 5 days, or 120 mg kg$^{-1}$ gemcitabine once every 3 days for 3 weeks; some mice were administered two cycles of dosing regimen with 2 weeks off between daily schedules and 1 week off for the intermittent schedule. Cytarabine and gemcitabine were formulated in PBS; cabazitaxel was made fresh daily, formulated in 5% ethanol/5% Tween-80/5% glucose water; all treatments were administered by intraperitoneal injection. Ruxolitinib was reconstituted in 0.5% methylcellulose with overnight end-over-end mixing and administered by oral gavage twice daily for 5 days for 4 weeks. Treatment started on day 10 (CHRF288-11-LUC/YFP+) or 17 (ML-2-LUC/YFP+) when significant bone marrow engraftment was observed, as determined by imaging analysis. Similarly, treatment started on day 13 for mice primary recipients with Mll$^{PTD/wt}$:Flt3$^{ITD/ITD}$ dKI AML or day 7 for secondary recipients, and day 17 or 20 for CG-V617-Luc+ AMKL after first observation of >2% Ly5.2+ or GFP/mCherry-double+ cells in peripheral blood or spleen. Treatment started on day 11 for the AMKL PDX after first observation of >2% human CD45 cells in peripheral blood.

Bone marrow and splenocytes were isolated from mice transplanted with dKI AML and CG-V617 AMKL. Following red blood cells lysis, dKI AML leukemic blasts from vehicle treated mice were evaluated for accumulation of cytarabine and gemcitabine using the uptake assay described above. CG-V617 AMKL leukemic blasts from treatment-naive tertiary transplants were subjected to red blood cell lysis then subjected to cell viability assay using increasing concentrations of ruxolitinib as described above.

**Pharmacokinetics of ruxolitinib**. Animal studies were performed under protocols approved by the Institutional Animal Care and Use Committee at OSU. Single dose pharmacokinetic studies were conducted in 8–12-week-old female and male NSG and BoyJ mice to determine a ruxolitinib dose that produces human equivalent exposure. For these studies, 60 mg kg$^{-1}$ of ruxolitinib was administered by oral gavage in 0.5% methylcellulose. Serial blood collection was performed from 15 min up to 240 min after dose administration. Total ruxolitinib concentrations in plasma were measured using a modification of previously published methods[61]. Quantitation was carried out by liquid chromatography-tandem mass spectrometry with a Vanquish UHPLC system and a TSQ Quantum mass spectrometer (ThermoFisher Scientific); separation was achieved in 5 min using an Accucore Vanquish C18 column and the system was controlled using Thermo Trace Finder General Quan software.

**Immunoblot analysis**. For protein expression of p-STAT5A and total-STAT5A, cells were lysed and whole cell lysates were prepared using radio-immunoprecipitation assay buffer (Sigma-Aldrich) supplemented with protease and phosphatase inhibitors. Total cell lysate (20 µg) was separated by SDS-polyacrylamide gel electrophoresis according to the manufacturer's instructions (Life Technologies) and transferred to PVDF membranes. Western blot analysis was then performed using p-STAT5A (1:1000, Cell Signaling Technology, Danvers, MA) and t-STAT5A (1:1000, Cell Signaling Technology); secondary α-rabbit or α-mouse antibodies (1:2000, Jackson ImmunoResearch, West Grove, PA) were used and proteins were visualized using the SignalFire ECL Reagent (Cell Signaling Technology) on an Odyssey Fc Imaging System (LI-COR, Lincoln, NE). GAPDH (Santa Cruz Biotechnology, Inc., Dallas, TX) was used a loading control; immunoblots were performed a minimum of three times on samples collected from different experiments. Uncropped and unprocessed blots are provided in the Data Source file.

**Statistical analysis**. Prism software (GraphPad Software) was used for statistical analyses. Kaplan–Meier analysis of animal survival and statistical significance of data was determined by log rank test; $P < 0.05$ was considered statistically significant. All other statistical tests performed are indicated in corresponding text or figure legend; $P < 0.05$ was considered statistically significant. A linear regression

analysis was performed to evaluate a goodness of fit and determine the Pearson correlation between STAT5A expression and ruxolitinib sensitivity.

## Data availability
The RNA-seq data generated and analyzed in this study are available at the Gene Expression Omnibus (GEO) repository of the National Center for Biotechnology Information under accession code GSE126489. The authors declare that all data generated from this study are included in this publication and its Supplementary Information, Source Data file (Figs. 2–7; Supplementary Figs. 1, 3, 5, 7, 9, 13, 15, 16), or available from the corresponding author on request.

## Code availability
High-throughput screening data was analyzed using our in-house Robust Interpretation of Screening Experiments (RISE) application written in Pipeline Pilot (Accelrys, v8.5) and the R program (R Development Core Team). The Pipeline Pilot protocol (exported as a Pipeline Pilot formatted xml file), including the embedded R code, are reported in Supplementary Software.

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

## Acknowledgements

The authors thank Drs. Alex Sparreboom and Navjot Pabla for helpful discussions and insights and Dr. Adrianne M. Dorrance for providing critical reagents. This work was supported by the American Lebanese Syrian Associated Charities (ALSAC) of St. Jude Children's Research Hospital, OSU Comprehensive Cancer Center Pelotonia Funds, National Institutes of Health Cancer Center Support Grant P30 CA021765 (SJCRH), P30 CA016058 (OSUCCC), R01 CA138744 (S.D.B.), and F32 CA180513 (C.D.D.).

## Author contributions

C.D.D., T.A.G., R.K.G., and S.D.B. conceived the project. C.D.D., J.D., and A.C. performed mouse experiments. A.S. and C.D.D. performed high-throughput screening analyses. S.J.O. conducted ex vivo primary patient experiments. M.L., J.Y.J., S.H., D.R.B., and M.P. conducted cell biology experiments. Q.F. developed and analyzed the ruxolitinib pharmacokinetic assay. H.I., R.C.R., J.E.R., and T.A.G. provided clinical expertise. C.D.D. wrote the manuscript; all authors reviewed and edited the manuscript.

## Additional information

**Competing interests:** The authors declare no competing interests.

