## [Peer Review File · Nature Communications]

Reviewers' comments:

Reviewer #1 (Remarks to the Author):

This article was a pleasure to read, is impactful and well-designed, and I think it can be published in Nature Communications without modifications. The authors screened a collection of small molecules (including drugs and experimental agents) against a panel of pediatric AML cell lines. Confirmation of activity in primary patient lines, mechanistic assessment, and in vitro follow-up of several agents (gemcitabine, cabazitaxel, JAK), and the use of multiple engineered genetic AML models validated the potential for these screening hits to be contemplated for clinical trials.

The authors could consider expanding the mouse efficacy data figure panels in the body of the manuscript to encompass more than just the Kaplan-Meier curves - even if it looks 'less clean'. - for example Figure S11 would greatly add as the actual read-out of efficacy (survival being the outcome).

Bottom P8: should reference to Figure 5C actually be Figure 4C?

This is the shortest review I've ever written!

Reviewer #2 (Remarks to the Author):

Baker and coworkers describe a screen of multiple AML (various subtype) cell lines versus a large collection of bioactive small molecules including approved and investigation drugs.

They utilize the outcomes of these screens and their follow-up studies to suggest new therapeutic approaches for the treatment of AML (broadly) and a selected subtype (AMKL).

The data presented (the screen and follow-up studies) is solid (although several of the figures are difficult to visualize).

These types of studies are always of interest provided the cell models are deemed good facsimiles of in situ disease.

Where the work is lacking is a clear demonstration of mechanistic drivers of the outcomes which would drive new basic-research studies of AML or potential translation into clinical evaluation. Mechanistic advances are needed for publication of in Nature Comm.

The concept that increase of cellular uptake of gemcitabine is responsible for its increase cellular toxicity is very interesting. But to fully appreciate this as a driver of divergent toxicity their would need to be additional studies.

Can dose elevation of cytarabine produce equivalent cellular exposure and dose this then equate to similar cellular toxicity?

Does knockdown of key transporters or the enzymes responsible for in situ activation of these drugs alter their cytotoxicity profile?

Do the drugs stop the cell cycle at the same phase? Does gH2AX staining track with exposure or dose?

Most importantly - dose the divergent uptake of these drugs in vivo? Is there any evidence from published clinical work on these drugs that would support this theory as a driver of divergent clinical outcomes?

Local exposure and cellular uptake following systemic administration is difficult to track (and nearly impossible to control) in vivo. So is this only an interesting observation in cell culture?

In the second example - the taxane class of drugs is generally toxic to cells. And the taxanes are known to be difficult drugs to tolerate. So the authors need to make a strong case for why these drugs have to be considered. Are they so remarkably more active than others that emerged from the screen that they can't be overlooked? If so, is there a mechanistic reason for their superior activity in AML?

The same issue resolves around JAK inhibitors in AMKL. It's very possibly a nice outcome - but the mechanistic driver of this activity is not explored. And the in vivo outcome at 60 mg/kg BID is above the recommended dose in humans of 10 mg/kg (after allometric scaling). That could be overcome - but only if there were very strong mechanistic validation that the study should be done and doses potentially elevated alongside a validated PD marker.

Overall this is a good study but strong mechanistic lessons are typically needed to justify publication in Nature Comm.

Reviewer #3 (Remarks to the Author):

The manuscript by Drenberg et al. describes results of compound screens and subsequent drug evaluation studies aimed at identifying and validating effective therapeutics for the treatment of pediatric high-risk AML. To this end, the authors initially screen a library of 7389 compounds at a single concentration (10 μ M) and timepoint (72h) in a panel of 8 AML cell lines, which is particularly enriched for AMKL cell lines. They then select 458 compounds (including hits from the primary screen and "other compounds of interest") for a secondary screen at different doses in the same cell line panel, leading to the identification of 17 compounds with broad activity in all 8 cell lines. Among these, they select several compounds for validation studies in primary AML samples, xenograft and genetically engineered mouse models of AML. As main findings, the authors report that (1) the nucleoside analog gemcitabine has broad activity and is more effective than cytarabine (Ara-C), the current standard of care in AML; (2) cabazitaxel (and other taxanes) are broadly effective in AML; (3) JAK2 inhibitors (particularly ruxolitinib) show anti-leukemic effects in AMKL.

Overall, the study by Drenberg et al. takes on an important question - the search for more effective therapies in high-risk pediatric AML, and by focusing on clinically established agents pursues a rational approach that may lead to rapid clinical translation. The major findings, particularly the broad activity of gemcitabine and cabazitaxel in AML, are interesting and potentially important. However, neither validation studies in primary AML nor pre-clinical drug trials in animal models provide sufficient experimental support for these conclusions. Moreover, primary screening results and hit selection strategies are not presented in a clear and coherent way, mechanistic studies on gemcitabine and taxanes merely reproduce what is known about their basic mechanism of action without providing insight into mechanisms underlying hypersensitivity in AML, and the studies on JAK2 inhibitors seem preliminary and do not sufficiently establish these agents as AMKL therapeutics. Therefore, the manuscript in its current form does not provide a sufficient advance over current knowledge. For revising their manuscript, the authors should consider the following major concerns:

1. The presentation of primary screening design, results and hit selection strategies is insufficient.

Figure 1 merely provides a snapshot on applied strategies and integrated data, but does not enable the reader to extract actual results. Therefore, in addition to improving the description of screening methodology (in the main text and methods section), all primary screening data (of both primary and secondary screens) should be included as supplemental information. Hit selection strategies and the inclusion of additional compounds (in both secondary screens and validation studies) should be described more clearly.

2. The central conclusion that gemcitabine "...demonstrated very potent activity [...] in primary patient samples,..." (lines 107-108) is not sufficiently supported by data presented in Figure 2. While sensitivities in the 8 chosen AML cell lines (Figure 3a) looks quite impressive, such broad and pronounced sensitivity is not seen in primary AML (Figure 2). In fact, gemcitabine (and also cabazitaxel) show highly variable effects in the presented primary AML samples, and (in contrast to a major claim of this study) do not appear to be generally superior to cytarabine. Instead of just referencing Figure 2 without any conclusions (lines 100-102), the authors should discuss these findings in light of their main conclusions.

3. In addition to a lack of evidence validating the superiority of gemcitabine over cytarabine in primary AML, the presented in-vivo studies appear to suffer from substantial shortcomings in the experimental design. Based on previous studies, in which the authors observed pronounced toxicity of cytarabine in NSG mice, in all in-vivo experiments they use an unusually low dose of cytarabine (50 mg/kg every 4 days), which they determine simply by matching the identified tolerable dose of gemcitabine. This strategy ignores possible differences in basic pharmacodynamics/pharmacokinetics of the two agents, as well as their common dosing in previous mouse trials and in the clinics. While pronounced toxicity of cytarabine in NSG mice presents a challenge, it is completely unclear why the authors used the same low dose in syngeneic mouse models where higher dosed cytarabine regimens (that better mirror its clinical use) are well-established and tolerated (i.e. 50-100 mg/kg daily). Therefore, a key experiment to convincingly show superiority of gemcitabine over cytarabine would be to first determine the MTD of gemcitabine in syngeneic models, and then compare it to an established appropriate cytarabine regimen.

4. The finding that JAK2 inhibitors (particularly ruxolitinib) show selective activity in AMKL appear to be preliminary. The presented in-vivo studies (Figure 6d-f) all involve AMKL cases harboring JAK2 mutations, where some activity of ruxolitinib must be expected – and the observed effects are in fact quite disappointing. Studies in AMKL cell lines indicate that ruxolitinib has also activity in non-JAK2 mutant AMKL, but claiming that these effects are selective for AMKL would require studies on a larger cohort of non-AMKL leukemias, in-vivo studies in such JAK2 wildtype AMKL cases, and some mechanistic insight into the basis of these effects.

In addition to these major concerns, the following minor concerns should be addressed:

- related to point 4, while the major focus and findings of this study is on high-risk pediatric AML, the selection of studied cell lines is highly biased for AMKL, and the assignment of other AML cell lines to certain "subtypes" appears to be problematic. Specifically, the authors divide other AML lines into MLL-rearranged and FLT3-ITD+, completely ignoring the fact that both FLT3-ITD+ lines (MOLM13, MV4;11) also harbor an MLL rearrangement, which as a disease-defining mutation is highly relevant to their biology. This mis-assignment should be rectified (a study comparing just AMKL and MLL-rearranged seems a valid strategy)
- The description of in vivo efficacy studies of different compounds in different models is extremely repetitive; these data should be presented in more concise/integrated way.
- Line 109: „A panel of AML cell lines were..." should read "was"
- Line 140 „...over the 72 run..." unit missing
- Line 150 „...higher accumulation gemcitabine and..." should read "accumulation of gemcitabine"

- Line 176 wrong reference: Figure 5c should be Figure 4c
- Line 479 „...we successful established...” should read “successfully”
- Line 502 „...ruxolitinib selective...” should read “ruxolitinib’s”
- Line 765 „...and luminescent was...” should read “luminescence”
- Line 849 „_ENREF_57”
- Figure 1a) 8086 compounds here when there is only talk of 7389 in the corresponding text; clarify
- Figure 3a) cell lines not color-coded c) maybe have a legend next to the figure instead of in text
- Figure 4 „(B)” should stand before „CHRF288-11, ML-2, and MOLM-13...”; text indicates DAPI, but PE shown in plot
- Figure S6d) x axis label wrong (at least not same as in figures b and c)

Reviewers' comments:

Reviewer #1 (Remarks to the Author):

This article was a pleasure to read, is impactful and well-designed, and I think it can be published in Nature Communications without modifications. The authors screened a collection of small molecules (including drugs and experimental agents) against a panel of pediatric AML cell lines. Confirmation of activity in primary patient lines, mechanistic assessment, and in vitro follow-up of several agents (gemcitabine, cabazitaxel, JAK), and the use of multiple engineered genetic AML models validated the potential for these screening hits to be contemplated for clinical trials.

1. The authors could consider expanding the mouse efficacy data figure panels in the body of the manuscript to encompass more than just the Kaplan-Meier curves - even if it looks 'less clean'. - for example Figure S11 would greatly add as the actual read-out of efficacy (survival being the outcome).

Response: We appreciate the reviewer's comment; as suggested we have revised Figure 5 and added new Figure 7 to include *in vivo* bioluminescent imaging data or peripheral blood infiltration based on detection of human CD45, Ly5.2+, or GFP/mCherry-double positive cells depending on the model.

2. Bottom P8: should reference to Figure 5C actually be Figure 4C?

Response: This was an oversight, the text has been corrected.

This is the shortest review I've ever written!

Reviewer #2 (Remarks to the Author):

Baker and coworkers describe a screen of multiple AML (various subtype) cell lines versus a large collection of bioactive small molecules including approved and investigation drugs. They utilize the outcomes of these screens and their follow-up studies to suggest new therapeutic approaches for the treatment of AML (broadly) and a selected subtype (AMKL).

The data presented (the screen and follow-up studies) is solid (although several of the figures are difficult to visualize).

These types of studies are always of interest provided the cell models are deemed good facsimiles of in situ disease.

Where the work is lacking is a clear demonstration of mechanistic drivers of the outcomes which would drive new basic-research studies of AML or potential translation into clinical evaluation. Mechanistic advances are needed for publication of in Nature Comm.

The concept that increase of cellular uptake of gemcitabine is responsible for its increase cellular toxicity is very interesting. But to fully appreciate this as a driver of divergent toxicity their would need to be additional studies.

1. Can dose elevation of cytarabine produce equivalent cellular exposure and dose this then equate to similar cellular toxicity?

Response: We appreciate the reviewers comment. To address this, we have performed uptake assays using 10 and 100 μ M Ara-C and the data are included in revised Figure 3 panels f and g. The manuscript results have been revised (page 9 line 182) as follows: “Next, we performed uptake assays at higher concentrations of cytarabine (10 and 100 μ M) to determine if dose escalation could result in equivalent intracellular exposure. In a comparative analysis after 5 minute exposure to cytarabine (10 or 100 μ M) we detected a significantly greater amount (1.6 to 3.7-fold and 7 to 12-fold, respectively) in all cell lines compared to 1 μ M cytarabine (**Figure 3f**). Furthermore, we found a 10-fold higher concentration of cytarabine produced nearly equivalent intracellular exposure to 1 μ M gemcitabine in all cell lines evaluated. Since transporters are biophysically complex and the interpretation of Michaelis-Menten parameters with regard to underlying transporter dynamics can be difficult²¹⁻²³, we evaluated transporter kinetics by calculating the transport efficiency for these compounds to gain a greater mechanistic understanding of these data. We found the overall transport of efficiency of gemcitabine to be significantly greater in AML cells versus cytarabine and this could not be overcome with increasing concentrations (**Figure 3g**).”

The discussion has been revised (page 23 line 517) as follows: “To that end, we performed uptake assays at increased concentrations of cytarabine and demonstrated that a 10-fold higher was necessary to achieve similar intracellular exposure compared to gemcitabine. These results are consistent with a previous report by Hertel et. al.⁶⁰, which demonstrated a minimum effective concentration of cytarabine is 10-fold higher than gemcitabine in CCRF-CEM cells (T lymphoblastoid cell line) and our own data showing a higher IC50 concentration for cytarabine compared to gemcitabine. Here, we show the transport efficiency of both drugs and show that even at the elevated concentrations of cytarabine, AML cells have the capacity to transport gemcitabine more efficiently which contributes to the observed differences in intracellular accumulation and greater cytotoxicity of gemcitabine.”

2. Does knockdown of key transporters or the enzymes responsible for in situ activation of these drugs alter their cytotoxicity profile?

Response: We appreciate the reviewers comment. To address this, we have performed knockdown experiments using siRNA targeting two key transporters OCTN1 and ENT1; this was followed by assessment of cytotoxicity. The data are included in revised Figure 3 panel h and i. The manuscript results have been revised (page 9 line 198) as follows: Given that previous investigations demonstrated that low cytarabine uptake in AML cells predicts poor response to therapy¹⁴ and taken our data showing uptake and transport efficiency as a major contributor to the enhanced sensitivity of AML cells to gemcitabine

we performed knockdown experiments targeting two key transporters involved in nucleoside uptake OCTN1 and ENT1.^{17,24} Using siRNA we were able to achieve a 54% and 41% reduction in expression of OCTN1 and ENT1, respectively (**Figure 3h**). At 24 h post-transfection we treated cells transfected with siRNA-Control, siRNA-OCTN1, and siRNA-ENT1 with PBS, gemcitabine (10 nM), or cytarabine (100 nM) for 16 h then assessed for alterations in their cytotoxicity profile. Since cytarabine and gemcitabine are S-phase specific nucleoside analogs, we evaluated alterations in cell cycle as the read-out of cytotoxicity. Inhibition of both transporters resulted in a decrease of cytarabine- and gemcitabine-mediated accumulation of cells in S-phase; the effect was greater with gemcitabine in ENT1 deficient cells and comparable in both OCTN1 and ENT1 deficient cells with cytarabine (**Figure 3i**).

The discussion has been revised (page 24 line 539) as follows: "This is consistent with our previous report showing over-expression of OCTN1 in HEK293 cells resulted in increased sensitivity to multiple nucleoside analogs, including cytarabine¹⁷ and our current findings that inhibition of OCTN1 and ENT1 reduced the accumulation of AML cells in S-phase with gemcitabine or cytarabine treatment."

3. Do the drugs stop the cell cycle at the same phase? Does gH2AX staining track with exposure or dose?

Response: It has been well established that as a class, nucleoside analogues like Ara-C and gemcitabine, block cells at the S phase of the cell cycle (Hertel et al., Cancer Res 1990; Shi et al., Cancer Res 2001; Zhu et al., JBC 2000 PMID 10827186) and induce gH2AX (Bonner et al., Nat Rev Cancer 2008 PMID 19005492; Ewald et al., Oncogene 2008 PMID 18955977). As indicated in response to the Reviewer's previous comment and consistent with the literature, we show accumulation of cells in S-phase with treatment of Ara-C and gemcitabine in siRNA Control cells (without inhibition of transporters) and the data are shown in Revised Figure 3 panel i.

4. Most importantly - dose the divergent uptake of these drugs in vivo? Is there any evidence from published clinical work on these drugs that would support this theory as a driver of divergent clinical outcomes?

Response: Clinical importance of the accumulation and retention of activate Ara-C triphosphate metabolite has been well established (Ref Rustum et al., Proc AACR 1976; Rustum et al., Cancer Res 1979; Plunkett et al., Semin Oncol 1985) and the therapeutic effect of Ara-C has been shown to be greater in leukemia blasts that have long retention times of intracellular Ara-CTP during therapy (Plunkett et al., Semin Oncol 1985). Further, the use of high-dose Ara-C in leukemia patients refractory to Ara-C implies that resistance may be overcome by increased intracellular triphosphate levels (Herzig et al., Blood 1981). This is also supported by our own report (Drenberg et al., Ca Res), showing low expression of OCTN1 strongly predicts poor event-free survival and overall survival in multiple cohorts of pediatric and adult AML patients. Likewise, the absence of ENT1 is associated with reduced survival in patients with gemcitabine-treated pancreatic cancer (Spratlin et al., CCR 2004). Additionally, ENT1 protein levels have been shown to predict

response to gemcitabine in patients with pancreatic cancer (Farrell et al., Gastroenterology 2009 PMID 18992248)

5. Local exposure and cellular uptake following systemic administration is difficult to track (and nearly impossible to control) in vivo. So is this only an interesting observation in cell culture?

Response: From a pharmacokinetic perspective gemcitabine exhibits more favorable attributes including slower cellular elimination half-life exhibiting both monophasic and biphasic properties. At higher cellular concentrations gemcitabine triphosphate exhibits more biphasic elimination with a prolonged half-life of 15-24 hours as compared with monophasic elimination of 4-6 hours in lower cellular concentrations. This is a unique property of gemcitabine that aids in the self-potentiating mechanism by reducing elimination and further promoting the accumulation of the active metabolite (blocks monophosphate deamination). In contrast, Ara-C has more rapid triphosphate elimination with a biphasic elimination half-life with α half-life 7-20 min and β half-life 2-3h (Wang et al., Biol Blood Marrow Transplant 2014 PMID; Mini et al., Ann Oncol 2006 PMID 16807468; Nieto et al., Biol Blood Marrow Transplant 2012 PMID 22643322; Shanks et al., Clin Cancer Res 2005 PMID 15930361; Plunkett et al., Semin Oncol 1996 PMID 8893876; Sampath et al., Oncogene 2003 PMID 14663485; Hamada et al., Clin Pharmacokinet 2002 PMID 1216278; Cheson BD, Keating MJ, Plunkett W. Nucleoside analogs in cancer therapy. New York: Marcel Dekker, Inc. 1997. Chapter 1: Nucleoside analogs: cellular pharmacology, mechanisms of action, and strategies for combination therapy and Chapter 3: Pharmacokinetics of purine nucleoside analogs; Gandhi and Plunkett Clin Pharmacokinet 2002 PMID 11888330)

6. In the second example - the taxane class of drugs is generally toxic to cells. And the taxanes are known to be difficult drugs to tolerate. So the authors need to make a strong case for why these drugs have to be considered. Are they so remarkably more active than others that emerged from the screen that they can't be overlooked? If so, is there a mechanistic reason for their superior activity in AML?

Response: We appreciate the Reviewer's comment. Cabazitaxel, the most recently FDA approved taxane was one of the most potent agents with low nanomolar activity in all the cell lines screened (Table 1) and met our criteria for further evaluation: 1) the drug is FDA-approved, 2) a pediatric dose has been determined or the agent is in phase 1 pediatric testing, 3) the drug is not currently used clinically or under clinical investigation for the treatment of adult or pediatric AML. Taken together, these criterion permit rapid translation to a clinical trial in pediatric AML; this is essential as we currently have access to few if any investigational agents for this disease in the pediatric population. Cabazitaxel fulfilled this criteria and was selected among the other commercially available anti-microtubule agents because it was the most potent and appears to be less neurotoxic than other tubulin poisons. In addition, the combination of a taxane with gemcitabine is a widely used standard of care regime or has shown promise to treat a variety of solid tumors including breast (Hu et al., Springerplus 2014), pancreatic (Goldstein et al., JNCI 2015 PMID 25638248), NSCLC (Esteban et al., Proc Am Soc Clin Oncol 1998), and most

notably pediatric refractory bone sarcoma (Navid et al., Cancer 2008 PMID 18484657; Song et al., *Pediatr Blood Cancer* 2014 PMID 24692087).

7. The same issue resolves around JAK inhibitors in AMKL. Its very possibly a nice outcome - but the mechanistic driver of this activity is not explored. And the in vivo outcome at 60 mg/kg BID is above the recommended dose in humans of 10 mg/kg (after allometric scaling). That could be overcome - but only if there were very strong mechanistic validation that the study should be done and doses potentially elevated alongside a validated PD marker.

Response: We appreciate the Reviewer's comment. In the revised manuscript, we have incorporated RNAseq data showing that the expression levels of multiple JAK-STAT family members are significantly increased in AMKL cell lines compared to non-AMKL and performed multiple in vitro assays to investigate the underlying mechanism and the data are included in revised Figure 6 panels b-h. The manuscript results have been revised (page 18 line 389) as follows: Next, we compared the expression levels of JAK-STAT family members among a panel of AMKL (JAK mutated: CHRF288-11, CMK, CMY; CBFA2T3-GLIS2+: CMS, M07e, M-MOK, WSU) and non-AMKL (MLLr: ML-2; MLLr with FLT3-ITD: MOLM-13, MV4-11; other: U937) cell lines using RNAseq data to provide insight into the mechanism driving the subtype selective activity of JAKis. We found multiple JAK-STAT family members including JAK1 (2-fold, $P=0.003$), JAK3 (9.3-fold, $P=0.003$), STAT3 (3.5-fold, $P=0.003$), STAT5A (6-fold, $P=0.003$), STAT5B (2.9-fold, $P=0.009$) are significantly increased in AMKL cell lines compared to the non-AMKL (**Figure 6b**). We validated our findings using an expanded panel of AML cell lines and confirmed STAT5A expression is significantly higher at both the RNA and protein level in AMKL compared to non-AMKL, with the exception of HEL cells that harbor a JAK2V617F mutation and served as a positive control (**Figure 6c and e**). Furthermore, we determined there is a high and significant ($R^2=0.76$, $P=0.01$) correlation between STAT5A expression and ruxolitinib sensitivity (IC_{50}) in the AMKL cell lines (**Figure 6d**); the sensitivity to ruxolitinib was also associated with a reduction in p-STAT5A signaling in AMKL regardless of JAK mutation status (**Figure 6f**). Previous investigations have shown JAK signaling can be stimulated by a variety of cytokines/chemokines including BMP2⁴¹, erythropoietin (EPO)⁴², and thrombopoietin (TPO).^{43,44} Since TPO plays a primary role and EPO, to a lesser extent, in the maintenance of megakaryocytes, and given our previous report showing elevated BMP2 expression in CBFA2T3-GLIS2-positive AMKL patients²⁸ we determined if co-exposure would affect ruxolitinib sensitivity. Therefore, we performed a cell viability assay using the CHRF288-11 AMKL cell line and increasing concentrations of ruxolitinib with or without 100 ng/mL of BMP2, EPO, or TPO. We found that TPO, a glycoprotein critical to megakaryocyte differentiation, caused a rightward shift of the dose-response curve resulting in a higher IC_{50} concentration (119 nM to 480 nM) and an indication of JAKi resistance (**Figure 6g**). Moreover, the observed resistance correlated with an induction of pSTAT5A signaling following stimulation with TPO (**Figure 6h**).

The discussion has been revised (page 28 line 613) as follows: Interestingly, we found a significantly higher expression of STAT5A in AMKL compared to non-AMKL cell lines, including those with FLT3-ITD; and expression highly correlated with ruxolitinib sensitivity

compared to other JAK-STAT family members. Our findings suggest that targeting STAT5A is integral to the underlying mechanism driving ruxolitinib's selective activity in AMKL.

The *in vivo* dose was selected based on previous publications that treated mouse models of MPN with 60 up to 180mg/kg (Koppikar et al., Blood 2010 115(14):2919 PMID 20154217; Quintas-Cardama et al., Blood 2010 115(15):3109 PMID 20130243) and ETP leukemia with 60mg/kg twice a day (Treanor et al., JEM 2014 211(4):701 PMID 24687960); these doses were associated with inhibition of JAK-STAT signaling *in vivo* and improvements in hematological features or survival.

Overall this is a good study but strong mechanistic lessons are typically needed to justify publication in Nature Comm.

Reviewer #3 (Remarks to the Author):

The manuscript by Drenberg et al. describes results of compound screens and subsequent drug evaluation studies aimed at identifying and validating effective therapeutics for the treatment of pediatric high-risk AML. To this end, the authors initially screen a library of 7389 compounds at a single concentration (10 μ M) and timepoint (72h) in a panel of 8 AML cell lines, which is particularly enriched for AMKL cell lines. They then select 458 compounds (including hits from the primary screen and "other compounds of interest") for a secondary screen at different doses in the same cell line panel, leading to the identification of 17 compounds with broad activity in all 8 cell lines. Among these, they select several compounds for validation studies in primary AML samples, xenograft and genetically engineered mouse models of AML. As main findings, the authors report that (1) the nucleoside analog gemcitabine has broad activity and is more effective than cytarabine (Ara-C), the current standard of care in AML; (2) cabazitaxel (and other taxanes) are broadly effective in AML; (3) JAK2 inhibitors (particularly ruxolitinib) show anti-leukemic effects in AMKL.

Overall, the study by Drenberg et al. takes on an important question - the search for more effective therapies in high-risk pediatric AML, and by focusing on clinically established agents pursues a rational approach that may lead to rapid clinical translation. The major findings, particularly the broad activity of gemcitabine and cabazitaxel in AML, are interesting and potentially important. However, neither validation studies in primary AML nor preclinical drug trials in animal models provide sufficient experimental support for these conclusions. Moreover, primary screening results and hit selection strategies are not presented in a clear and coherent way, mechanistic studies on gemcitabine and taxanes merely reproduce what is known about their basic mechanism of action without providing insight into mechanisms underlying hypersensitivity in AML, and the studies on JAK2 inhibitors seem preliminary and do not sufficiently establish these agents as AMKL therapeutics. Therefore, the manuscript in its current form does not provide a sufficient advance over current knowledge. For revising their manuscript, the authors should consider the following major concerns:

1. The presentation of primary screening design, results and hit selection strategies is insufficient. Figure 1 merely provides a snapshot on applied strategies and integrated data, but does not enable

the reader to extract actual results. Therefore, in addition to improving the description of screening methodology (in the main text and methods section), all primary screening data (of both primary and secondary screens) should be included as supplemental information. Hit selection strategies and the inclusion of additional compounds (in both secondary screens and validation studies) should be described more clearly.

Response: We appreciate the Reviewer's comment. To comply with the Reviewer's recommendation, we have included new Supplemental Table S3 that indicates the percent inhibition of active compounds in the primary screen; we have also clarified our selection criteria. The manuscript results have been revised (page 5 line 90) as follows: "the activity of these compounds are reported in **Table S3**. A total of 458 compounds were tested in a secondary screen performed in a dose-response manner (10-point curve, 1 nM-10 μ M), including FDA approved compounds with inhibition >50% in more than one cell line in the primary screen, analogs of these hits, and other compounds of interest (e.g., NAMPT inhibitors) not included in the primary screen; clinical phase of testing was also taken into consideration."

The manuscript results have been revised (page 5 line 103) as follows: "Next, in a low-throughput manner using a panel of cell lines and primary patient samples we validated compounds from these drug classes and others targeting pathways known to be upregulated or mutated in MLLr or AMKL (eg. tremetinib, RAS pathway; alisertib, aurora kinase; RG7112, MDM2 inhibitor)⁷⁻⁹ (**Figure 2 and S3, Table S5**)."

The manuscript results have been revised (page 6 line 125) as follows: We used the following criteria for the advancement of compounds from the secondary screen that demonstrated broad activity (e.g. EC₅₀ < 1 μ M in all cell lines, Table 1) across subtypes: 1) the drug is FDA-approved, 2) a pediatric dose has been determined or the agent is in phase 1 pediatric testing, and 3) the drug is not currently used clinically or under investigation for the treatment of adult or pediatric AML. Taken together, these criterion permit rapid translation to a clinical trial in pediatric AML; this is essential as we currently have access to few if any investigational agents for this disease."

The manuscript methods have been revised (page 35 line 857) as follows: Compounds selection for the secondary screen was based on the following: 1) demonstration of >50% inhibition in one cell line in the primary screen; 2) currently or previously evaluated in clinical phase testing; 3) analogs of compounds in the primary screen showing activity that were not included; 4) compound classes of interest not included in primary screen.

The screening data for the secondary screen was previously included as Table S3; in the revised manuscript it is new **Supplemental Table S4**.

2. The central conclusion that gemcitabine "...demonstrated very potent activity [...] in primary patient samples,..." (lines 107-108) is not sufficiently support by data presented in Figure 2. While sensitivities in the 8 chosen AML cell lines (Figure 3a) looks quite impressive, such broad and pronounced sensitivity is not seen in primary AML (Figure 2). In fact, gemcitabine (and also cabazitaxel) show highly variable effects in the presented primary AML samples, and (in contrast

to a major claim of this study) do not appear to be generally superior to cytarabine. Instead of just referencing Figure 2 without any conclusions (lines 100-102), the authors should discuss these findings in light of their main conclusions.

Response: We appreciate the Reviewer's comment. In general, primary AML patient samples do not extensively proliferate *ex vivo* and maintenance of cell viability is a major challenge. We therefore conduct such experiments using stromal support cells to help maintain cell viability; this also challenges compounds being evaluated. This overt lack in cycling of AML patient samples in an *ex vivo* setting impact the activity of particular drug classes, especially those that work on S phase of the cell cycle like nucleoside analogues (as noted in a previous response to Reviewer #2). Therefore, we have toned down our statement regarding the activity of gemcitabine in primary samples and included supplemental data showing a lack in cycling of AML patient samples under the *ex vivo* conditions we used (new Supplemental Figure S4). The manuscript results have been revised (page 5 line 108) as follows: "For these assays, primary patient samples were co-cultured with mesenchymal stromal cells (MSCs), which secrete multiple cytokines that mimic the bone marrow microenvironment; this system gives stromal support to the primary blast samples while challenging the drug treatment. Cell viability and cell density were monitored every 24 h during the 96 h assay and are shown in **Figure S4**; these results demonstrate all primary samples experience a dramatic decrease in cell number at 24 h. While cell numbers are relatively maintained over the course of the assay only 1 of the primary samples actually doubles from 24 to 96 hours. This is an important observation especially in regard to drugs that specifically target the S phase cells and may contribute to modest activity of nucleoside analogues like cytarabine and gemcitabine in this assay. We observed the HDAC inhibitors panobinostat and romidepsin to have potent activity across subtypes; these findings are consistent with our previous report with panobinostat¹⁰ and support the ongoing clinical evaluation (NCT02676323) of this drug for pediatric AML. Similarly, the proteasome inhibitors, carfilzomib and bortezomib, demonstrated potent activity and have been extensively investigated in the clinic.¹¹"

The manuscript results have been revised (page 7 line 136) as follows: "...and taken these results were validated in a low-throughput manner (**Figure 3a**) and had comparable activity to cytarabine in primary patient samples (**Figure 2 and Table S5**), we selected this compound for further evaluation.

3. In addition to a lack of evidence validating the superiority of gemcitabine over cytarabine in primary AML, the presented in-vivo studies appear to suffer from substantial shortcomings in the experimental design. Based on previous studies, in which the authors observed pronounced toxicity of cytarabine in NSG mice, in all in-vivo experiments they use an unusually low dose of cytarabine (50 mg/kg every 4 days), which they determine simply by matching the identified tolerable dose of gemcitabine. This strategy ignores possible differences in basic pharmacodynamics/pharmacokinetics of the two agents, as well as their common dosing in previous mouse trials and in the clinics. While pronounced toxicity of cytarabine in NSG mice presents a challenge, it is completely unclear why the authors used the same low dose in syngeneic

mouse models where higher dosed cytarabine regimens (that better mirror its clinical use) are well-established and tolerated (i.e. 50-100 mg/kg daily). Therefore, a key experiment to convincingly show superiority of gemcitabine over cytarabine would be to first determine the MTD of gemcitabine in syngeneic models, and then compare it to an established an appropriate cytarabine regimen.

Response: We appreciate the Reviewer's insightful comment. In the revised manuscript, we have incorporated the requested information by establishing an MTD of daily (3mg/kg) and intermittent (120mg/kg q3d) administration of gemcitabine in immunocompetent mice and the results are shown in a new Supplemental Figure S12. We then performed efficacy studies to compare the established MTD of daily and intermittent gemcitabine to a MTD of Ara-C; the results are shown in a revised Figure 5 panels d and f. The manuscript results have been revised (page 14 line 310) as follows: "Next, we sought to evaluate efficacy in two syngeneic murine models using immunocompetent mice using a maximum tolerated dose of cytarabine that better reflects clinical regimens. First, we determined the MTD of gemcitabine in the Ly5.1 C57BL/6 (BoyJ) mouse strain. Tolerability of gemcitabine was performed using three doses (100, 120, 140 mg/kg) on an intermittent every three days for three weeks schedule and two doses (3, 6 mg/kg) on a daily for five days (**Figure S12**).^{35,36} For efficacy studies in CG-V617-Luc+ quaternary transplants, mice were treated with gemcitabine (3 mg/kg daily x 5 days or 120 mg/kg every 3 days for 3 weeks) or cytarabine (50 or 100 mg/kg daily x 5 days). We found that both dosing regimens of cytarabine provided a survival advantage compared to vehicle treated mice in CG-V617-Luc+ quaternary transplants (50mg/kg dose median survival 63 vs 59 days, $P=0.0091$; 100mg/kg dose median survival 80 vs 59 days, $P=0.0091$) (**Figure 5d**). However, gemcitabine provided the greatest survival advantage (gemcitabine 3 mg/kg median survival 91 vs 59 days, $P=0.0091$; gemcitabine 120 mg/kg median survival >125 vs 59 days, $P=0.004$) and significantly prolonged survival versus 100 mg/kg cytarabine (gemcitabine 120 mg/kg median survival >125 vs 80 days, $P=0.0027$). Furthermore, treatment with gemcitabine significantly inhibited tumor burden compared to cytarabine as indicated by decreased infiltration in the peripheral blood (day 61, $P=0.0068$) (**Figure 5d**).

Lastly, we evaluated efficacy using a $Mll^{PTD/wt}; Flt3^{ITD/ITD}$ double knock-in murine model.^{27,28} For these studies, primary transplants were treated with gemcitabine or cytarabine (50 mg/kg every 4 days for 3 weeks), or cabazitaxel (5 mg/kg every 4 days for 3 weeks); whereas, secondary transplants were treated with gemcitabine at the daily and intermittent MTD (3 mg/kg daily x 5 days or 120 mg/kg every 3 days for 3 weeks) or cytarabine (100 mg/kg daily x 5 days). In the primary transplants, cytarabine did not significantly prolong survival compared with vehicle (median survival 50 vs 47 days; $P=0.4$); whereas leukemic mice treated with cabazitaxel (median survival 60 vs 47 days, $P=0.011$) achieved a significant survival advantage (**Figures 5e and S13**). However, gemcitabine provided the greatest survival advantage and significantly prolonged survival compared to all other treatment regimens (median survival 69 vs 47 days, $P=0.0013$; median survival 69 vs 50 days, $P=0.024$; median survival 69 vs 60 days, $P=0.007$). In the secondary transplants, cytarabine significantly prolonged survival compared with vehicle (median survival 49 vs 39 days; $P=0.009$); similarly the daily MTD of gemcitabine significantly prolonged survival compared with vehicle (median survival 49 vs 39 days;

$P=0.009$) (**Figure 5f**). We found the intermittent MTD of gemcitabine was not well tolerated in this model and mice only received 4 doses on this treatment regimen (**Figure S15**); we attributed the lack of tolerability to the short time frame between treatment initiation and sublethal irradiation prior to transplant. Interestingly, however, the weights fully recovered and the mice had a profound response to the limited regimen and achieved a significant survival advantage compared to vehicle (median survival 52 vs 39 days, $P=0.0006$) (**Figure 5f**). All treatment groups had a significant reduction in tumor burden compared to vehicle on day 13 as indicated by decreased infiltration in the peripheral blood (cytarabine, $P=0.019$; gemcitabine daily MTD, $P=0.0046$ and intermittent MTD $P=0.011$)."

4. The finding that JAK2 inhibitors (particularly ruxolitinib) show selective activity in AMKL appear to be preliminary. The presented in-vivo studies (Figure 6d-f) all involve AMKL cases harboring JAK2 mutations, where some activity of ruxolitinib must be expected – and the observed effects are in fact quite disappointing. Studies in AMKL cell lines indicate that ruxolitinib has also activity in non-JAK2 mutant AMKL, but claiming that these effects are selective for AMKL would require studies on a larger cohort of non-AMKL leukemias, in vivo studies in such JAK2 wildtype AMKL cases, and some mechanistic insight into the basis of these effects.

Response: We appreciate the Reviewer's comment. Please refer to our previous response to Reviewer #2's comment 7 regarding mechanistic insight into the selective activity of JAK inhibitors; we have included additional data in revised Figure 6 panels b-h to show AMKL cells have significantly greater expression of STAT5 at both the RNA and protein level compared to non-AMKL cells regardless of JAK mutation status. We would like to clarify that the patient sample used to establish the PDX model harbored the CBFA2T3-GLIS2 fusion but does not carry a JAK2 mutation. Since our previous submission, we incorporated luciferase into the CBFA2T3-GLIS2/JAK2V617F-induced transduction/transplantation model to permit monitoring engraftment by bioluminescence imaging; the signal is observed in the hind limbs at an early time point post-transplant compared to detection of GFP/mCherry-double positive cells in the peripheral blood and allowed us to initiate dosing earlier; it also is important to mention that due to the severely enlarged spleens (weight >1g) in this model imaging is discontinued after day 45 and tumor burden is monitored by peripheral blood for the remainder of the study. Our previous study showed a significant yet modest 18 day survival advantage whereas in the luciferase+ model ruxolitinib prolonged survival 44 days longer than vehicle treated mice; the results are shown in new Figures 7 panel d and S8 panel e). The manuscript results have been revised (page 12 line 268) as follows: "To permit early monitoring of tumor burden by bioluminescence imaging, we transduced AMKL blasts isolated from secondary transplants with luciferase-BFP retrovirus. The bioluminescence signal could be detected in the hind limbs 15 days earlier than mCherry/GFP-double positive cells in the peripheral blood and CG/V617-luciferase-positive (CG/V617-Luc+) tertiary transplants exhibited a highly penetrant phenotypically similar AMKL to the CG/V617 transplant recipients (**Figure S8e**)."

In addition to these major concerns, the following minor concerns should be addressed:

- related to point 4, while the major focus and findings of this study is on high-risk pediatric AML, the selection of studied cell lines is highly biased for AMKL, and the assignment of other AML cell lines to certain “subtypes” appears to be problematic. Specifically, the authors divide other AML lines into MLL-rearranged and FLT3-ITD+, completely ignoring the fact that both FLT3-ITD+ lines (MOLM13, MV4;11) also harbor an MLL rearrangement, which as a disease-defining mutation is highly relevant to their biology. This mis-assignment should be rectified (a study comparing just AMKL and MLL-rearranged seems a valid strategy)

Response: We acknowledge and agree with the Reviewer’s comment. We have rectified this mis-assignment as suggested. The manuscript introduction has been revised (page 3, line 63) as follows: “Here, we report the results of a large-scale screen of human cancer cell lines representing two high-risk subtypes of pediatric AML including: *MLL* rearranged (MLLr) with or without a co-occurring *FLT3*-internal tandem duplication (FLT3-ITD) mutation and non-Down syndrome acute megakaryoblastic leukemia (AMKL).” The manuscript result have been revised (page 4 line 78) as follows: “...represent subtypes of pediatric AML (e.g., AMKL, FLT3-ITD+ with MLLr, MLLr) associated with a high risk of relapse...” The figure legend for Figure 1, now indicates the cell lines in blue are FLT3-ITD+ with MLLr.

- The description of in vivo efficacy studies of different compounds in different models is extremely repetitive; these data should be presented in more concise/integrated way.

Response: We appreciate the Reviewer’s comment and have revised the text describing the in vivo efficacy studies as suggested refer to page 12 line 291 through page 17 line 360.

- Line 109: „A panel of AML cell lines were...” should read “was”
- Line 140 „...over the 72 run...” unit missing
- Line 150 „...higher accumulation gemcitabine and...” should read “accumulation of gemcitabine”
- Line 176 wrong reference: Figure 5c should be Figure 4c
- Line 479 „...we successful established...” should read “successfully”
- Line 502 „...ruxolitinib selective...” should read “ruxolitinib’s”
- Line 765 „...and luminescent was...” should read “luminescence”
- Line 849 „_ENREF_57”

Response: all text has been corrected as suggested.

- Figure 1a) 8086 compounds here when there is only talk of 7389 in the corresponding text; clarify

Response: The number of compounds indicated in the text is correct; Figure 1 has been revised to indicate the correct number of compounds in the primary screen.

- Figure 3a) cell lines not color-coded c) maybe have a legend next to the figure instead of in Text

Response: This was an oversight, color coding of the cell lines has been added to Figure 3 panel a.

- Figure 4 „(B)“ should stand before „CHRF288-11, ML-2, and MOLM-13...“; text indicates DAPI, but PE shown in plot

Response: The text is correct, DAPI was used to determine alterations in cell cycle; the label in plots has been corrected to indicate Pac Blue which was the laser in the flow cytometer used to detect DAPI.

- Figure S6d) x axis label wrong (at least not same as in figures b and c)

Response: The x-axis label has been changed to be consistent with panels b and c (now Figure S7).

REVIEWERS' COMMENTS:

Reviewer #1 (Remarks to the Author):

The authors have responded adequately to my queries (Reviewer 1), and that appears to be the case for questions from the other reviewers also. Ready for publication.

Reviewer #2 (Remarks to the Author):

The revised manuscript has not changed significantly enough to warrant publication in Nature Comm.

Screens of approved and investigational drugs versus good cell models of disease are informative for 2 reasons. 1) they can be used as a systems biology experiment as they offer pharmacological insight into cellular mechanisms contributing to viability and pharmacology. 2) they can be used to justify re-purposing of a drug into clinical trials.

The authors pursue the 2nd track and highlight drugs which may be of use in human clinical trials of different subtypes of AML.

They offer 3 types of data. 1) the screen. 2) confirmation in cell lines. 3) in vivo data.

This is fine for publication in multiple journals but for publication in Nature Comm. the authors must demonstrate convincing mechanistic rational justifying their suggestions. The superiority of gemcitabine noted in the authors screens and in vivo is compelling. If this is predictive of superiority in human clinical trials the mechanistic reasons driving that superiority in cell and mouse models must translate to in situ human diseased cells. That is why detailed MOA studies are so crucial.

The previous reviews offered some suggestions - and the authors diligently performed several of the suggested experiments. But they missed the message: figure out why the actions of gemcitabine is superior. The same for cabazitaxel and ruxilotinib.

It should be noted, the authors should not attribute a 41-55% reduction in mRNA for a few transporters (without western blot confirmation of divergent protein expression) as a mechanistic proof that altered cellular uptake is the mechanistic key. Those targets should be CRISPR'd out and, if the cells remain viable, they should be used extensively in vitro and in vivo studies to unambiguously demonstrate the role of drug transport. Patient tissue needs to play a role as well. When MOA lessons are learned it needs to be clear that the cellular attributes driving the MOA-based actions of drugs in models is present in the human disease.

Reviewer #3 (Remarks to the Author):

In response to my questions and concerns, the authors have revised several statements in their manuscript and added a substantial amount of new experimental data in support of their main conclusions. Overall, I am very satisfied with their response. In-vivo treatment studies now clearly demonstrate that alternative chemotherapeutic regimens such as gemcitabine or cabazitaxel (both of which not used or even considered in AML) can be superior to Ara-C. This seems the most important point of the paper, as it puts into question the validity of one of the most well-established chemotherapy regimens used in the clinic and suggests that treatment success could be improved by revisiting the activity of well-established/conventional cancer therapeutics in indications that

they are not used in. While some questions remain unanswered, this is a comprehensive study providing an impressive amount of pre-clinical drug testing data that will be of interest to the broader cancer research community (even beyond AML).

The only remaining request I have is that the authors should provide all primary data of primary and secondary drug screens (including non-scoring compounds; as requested in my previous comments). This should be feasible in a simple supplementary table and will be of great value for interested readers.

REVIEWERS' COMMENTS:

Reviewer #1 (Remarks to the Author):

The authors have responded adequately to my queries (Reviewer 1), and that appears to be the case for questions from the other reviewers also. Ready for publication.

Response: We thank the Reviewer for this comment.

Reviewer #2 (Remarks to the Author):

The revised manuscript has not changed significantly enough to warrant publication in Nature Comm.

Screens of approved and investigational drugs versus good cell models of disease are informative for 2 reasons. 1) they can be used as a systems biology experiment as they offer pharmacological insight into cellular mechanisms contributing to viability and pharmacology. 2) they can be used to justify re-purposing of a drug into clinical trials.

The authors pursue the 2nd track and highlight drugs which may be of use in human clinical trials of different subtypes of AML.

They offer 3 types of data. 1) the screen. 2) confirmation in cell lines. 3) in vivo data.

This is fine for publication in multiple journals but for publication in Nature Comm. the authors must demonstrate convincing mechanistic rational justifying their suggestions. The superiority of gemcitabine noted in the authors screens and in vivo is compelling. If this is predictive of superiority in human clinical trials the mechanistic reasons driving that superiority in cell and mouse models must translate to in situ human diseased cells. That is why detailed MOA studies are so crucial.

The previous reviews offered some suggestions - and the authors diligently performed several of the suggested experiments. But they missed the message: figure out why the actions of gemcitabine is superior. The same for cabazitaxel and ruxilotinib.

It should be noted, the authors should not attribute a 41-55% reduction in mRNA for a few transporters (without western blot confirmation of divergent protein expression) as a mechanistic proof that altered cellular uptake is the mechanistic key. Those targets should be CRISPR'd out and, if the cells remain viable, they should be used extensively in vitro and in vivo studies to unambiguously demonstrate the role of drug transport. Patient tissue needs to play a role as well. When MOA lessons are learned it needs to be clear that the cellular attributes driving the MOA-based actions of drugs in models is present in the human disease.

Response: We appreciate the Reviewer's comment. Thus, while we agree that a more in depth investigation into the mechanism of action is worthy for each of the three compounds highlighted in this manuscript; overall, we feel this is a comprehensive report and such studies will be pursued once additional funding has been secured.

Reviewer #3 (Remarks to the Author):

In response to my questions and concerns, the authors have revised several statements in their manuscript and added a substantial amount of new experimental data in support of their main conclusions. Overall, I am very satisfied with their response. In-vivo treatment studies now clearly demonstrate that alternative chemotherapeutic regimens such as gemcitabine or cabazitaxel (both of which not used or even considered in AML) can be superior to Ara-C. This seems the most important point of the paper, as it puts into question the validity of one of the most well-established chemotherapy regimens used in the clinic and suggests that treatment success could be improved by revisiting the activity of well-established/conventional cancer therapeutics in indications that they are not used in. While some questions remain unanswered, this is a comprehensive study providing an impressive amount of pre-clinical drug testing data that will be of interest to the broader cancer research community (even beyond AML).

The only remaining request I have is that the authors should provide all primary data of primary and secondary drug screens (including non-scoring compounds; as requested in my previous comments). This should be feasible in a simple supplementary table and will be of great value for interested readers.

Response: We appreciate the Reviewer's comment. As requested, we have included all of the data from the primary screen in new Supplementary Data 1. The entire data set from the secondary screen was previously provided as a Supplementary Table 4 and can now be found in new Supplementary Data 2.